# Building a framework for fake news detection in the health domain

**Juan R. Martinez-Rico** [1]*, **Lourdes Araujo**[1,2], **Juan Martinez-Romo**[1,2]

**1** NLP & IR Group, Dpto. Lenguajes y Sistemas Informáticos, Universidad Nacional de Educación a Distancia (UNED), Madrid, Spain, **2** Instituto Mixto de Investigación - Escuela Nacional de Sanidad (IMIENS), Madrid, Spain

* jrmartinezrico@invi.uned.es

**Data Availability Statement:** The source code associated with this article is available in the repositories https://github.com/jrmtnez/hnfc-agent and https://github.com/jrmtnez/hnfc-site. The dataset, composed of the URLs that allow access

## Abstract

Disinformation in the medical field is a growing problem that carries a significant risk. Therefore, it is crucial to detect and combat it effectively. In this article, we provide three elements to aid in this fight: 1) a new framework that collects health-related articles from verification entities and facilitates their check-worthiness and fact-checking annotation at the sentence level; 2) a corpus generated using this framework, composed of 10335 sentences annotated in these two concepts and grouped into 327 articles, which we call KEANE (faKe nEws At seNtence lEvel); and 3) a new model for verifying fake news that combines specific identifiers of the medical domain with triplets subject-predicate-object, using Transformers and feedforward neural networks at the sentence level. This model predicts the fact-checking of sentences and evaluates the veracity of the entire article. After training this model on our corpus, we achieved remarkable results in the binary classification of sentences (check-worthiness F1: 0.749, fact-checking F1: 0.698) and in the final classification of complete articles (F1: 0.703). We also tested its performance against another public dataset and found that it performed better than most systems evaluated on that dataset. Moreover, the corpus we provide differs from other existing corpora in its duality of sentence-article annotation, which can provide an additional level of justification of the prediction of truth or untruth made by the model.

## 1 Introduction

The use of disinformation has historically been employed by actors of all kinds to serve their purposes, be it political, strategic, economic, or ideological. However, with the advent of the internet and social media, the proliferation of this phenomenon has grown exponentially. Although the classic arena in which such processes usually occur is that of politics and international relations, the increase of disinformation and fake news in the field of healthcare has recently gained special attention due to the last pandemic. This is particularly significant because many patients and individuals often turn to these media sources for information regarding diseases, treatments, and medications, which can pose serious risks to their health [1]. According to [2] 58.5% of U.S. adults look for health information online and 35% use it to self-diagnose [3], avoiding going to a doctor. This has an impact on public health since 40% of the links shared on social networks related to these topics can be considered fake news [4].

to the original news items, and for each of them, the sentence-level annotation of both: fact-checking and check-worthines is available at the link https://doi.org/10.5281/zenodo.10802196.

**Funding:** INITIALS: LAS, JMR GRANT NUMBER: PID2019-106942RB-C32 FUNDER: Spanish Ministry of Science and Innovation URL FUNDER: https://www.ciencia.gob.es/ INITIALS: JMR, LAS GRANT NUMBER: TED2021-130398B-C21 FUNDER: Spanish Ministry of Science and Innovation URL FUNDER: https://www.ciencia.gob.es/ INITIALS: JMR, LAS GRANT NUMBER: PID2022-136522OB-C21 FUNDER: Spanish Ministry of Science and Innovation URL FUNDER: https://www.ciencia.gob.es/ INITIALS: LAS GRANT NUMBER: RAICES (IMIENS 2022) FUNDER: IMIENS (Instituto Mixto de Investigación-Escuela Nacional de Sanidad) URL FUNDER: https://www.imiens.es/.

**Competing interests:** The authors have declared that no competing interests exist.

To mitigate this problem as much as possible, a series of entities have emerged in several countries, usually managed by journalists and information professionals [5]. These entities are dedicated to evaluate the news and claims that are published in social networks and sensationalist or not too reliable media, pointing out which can be considered as false claims or fake news, and providing the evidence found to reach that conclusion. However, the immense amount of information generated every day by our society exceeds the control and evaluation capabilities of these entities, so automated systems are necessary to complement and/or replace in some cases the work they do.

One way to tackle this task is by using machine learning and natural language processing techniques. Nevertheless, it is not easy to address it due to several reasons: 1) there are a large number of claims to verify that spread quickly and in many cases their validity is temporary; for example, what is true today regarding a vaccination schedule tomorrow may be false, 2) in many cases this type of news are written to deceive the recipient, so it is even difficult for humans to distinguish between true and false news, and 3) machine learning methods are based on the existence of annotated examples with which to train the models but these resources are scarce, especially in specific domains.

Although generic methods of detecting fake news are applicable in the health field, the use of a characteristic language, the existence of specific resources such as knowledge bases or terminologies, and the fact of working in a restricted domain, should enable us to design more efficient models to carry out this task. Health disinformation use different writing styles depending on whether they are websites, health forums, or social media. It contains many medical terms and acronyms. It spread quickly due to its sensationalist and alarming nature. Due to the damage that this type of news can cause, a greater justification and interpretation of the results offered by detection systems is necessary. To that end, in this work we have addressed the issue of fake news, starting by generating a methodology that allows news to be collected along with its indications of veracity to facilitate the creation of a reference collection. This methodology has been established as a pipeline with a high degree of automation that would require minimal intervention to add new items that are continuously being captured from the Internet. With this collection of news, we will be able to evaluate the sentences included in these documents according to its check-worthiness (the identification of the statements that are worth checking) and truth value, and assign a final classification for every item to incorporate it into the dataset (Fig 1). To do this, new models have been proposed with

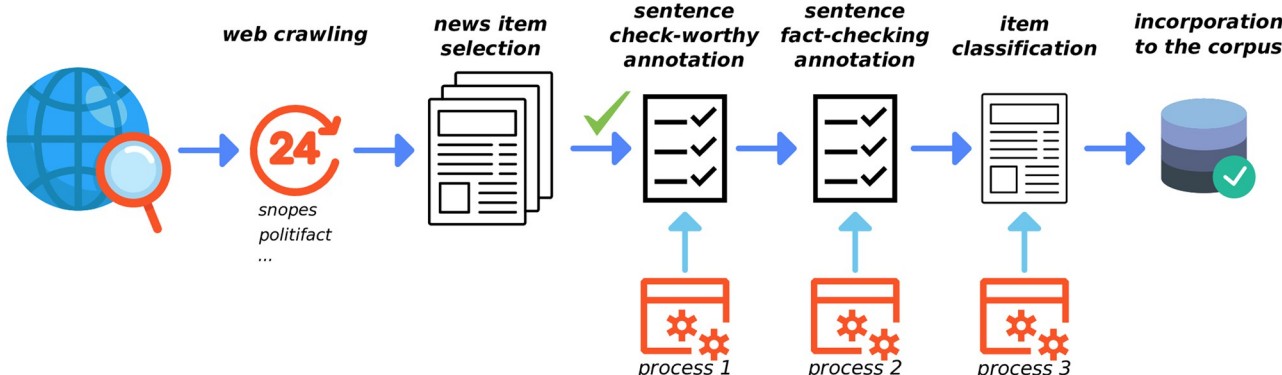

**Fig 1. Pipeline overview.** Verification sites are consulted every 24 hours. After a manual selection of the captured articles, a first process proposes a check-worthy classification that the user validates. Then, a second process proposes a fact-checking classification that the user validates again. Finally, a final process determines the veracity of the entire article based on the veracity of its sentences, and it is incorporated into the corpus.

state-of-the-art results that exploit syntactic and medical information and that allow justifying the reasons for having considered a news item as false or true.

Following the mentioned methodology, we have recovered a corpus of reference that will not only allow us to train and validate fake-news detection systems, but also ensures its continuous expansion. We have called it *KEANE*, which stands for faKe nEws At seNtence lEvel. This collection of news has also allowed us to design different approaches in the processes of extraction of features and classification that occur in the different stages of the pipeline. Among them we can highlight the use of ensembles of classifiers that simultaneously incorporate knowledge-based characteristics such as the subject-predicate-object triplets and information from the medical domain in the form of standardized identifiers of concepts.

We have organized the rest of the article as follows: Section 2 briefly reviews the different approaches made to the problem of detecting fake news and the development of datasets focused on this task, Section 3 presents the formulation of the problem and associated definitions, Section 4 sets out our research objective and contributions, Section 5 explains our methodology for building the dataset, the different models used in it, as well as the main statistics of the collected dataset, Section 6 shows the results obtained in the different stages of the pipeline, Section 7 discusses these results, and finally in Section 8 we draw the conclusions and possible directions for future work.

## 2 Related work

In this section we review previous works related to the detection of fake news. There are other lines of research mainly based on the analysis of the spread of disinformation on social networks, which go beyond detection and propose mitigation strategies for the effects of this disinformation [6, 7] or the immunization of the nodes causing it [8, 9]. In this work, however, we focus on detection. We divide detection approaches into two large groups: those that contemplate the article as a whole, and those that analyze their claims to determine the truth value of them. We also show different examples of datasets oriented to this task that have appeared over time.

### 2.1 Fake news detection methods

Identifying fake news is a truly complex task that has been addressed through different strategies over the last few years. To determine whether a news item is true or false, we must decide which type of features and which detection method we are going to use [10]. There are features that are inherent to the entire document, such as stylometric features that may indicate the presence of misleading content, and features based on determining the veracity of a claim. In this second case we can deduce the veracity of a document based on the veracity of the claims it contains, although if the document is a tweet or post on another social network, it may only contain a single statement. The latter approach implies dividing the task of detecting fake news into two main subtasks [11]: the identification of the statements that are worth checking (check-worthiness), and the determination of the veracity of the selected statements (fact-checking). Table 1 summarizes the different proposals detailed below.

**2.1.1 Document based fake news detection.**   Within the detection of fake news using the full article as a unit to be evaluated, we have grouped the existing methods into two main strategies.

*Document content based.* The first strategy considers only the textual content of the article by trying to associate misleading content with certain writing styles [12], using features such as bag-of-words vectors, part-of-speech tags, or Probabilistic Context Free Grammars to carry out this task. Text analysis techniques such as LIWC [13], and discourse level features that

Table 1. Fake news detection approaches.

| Method | Ref. | Document | | Claim | |
|---|---|---|---|---|---|
| | | Content | Context | Structured | Unstr. |
| Writting styles | [12] | ✓ | | | |
| Text analysis | [13] | ✓ | | | |
| Discourse analysis | [14] | ✓ | | | |
| Latent features (Transformers) | [15] | ✓ | | | |
| Embeddings | [16] | ✓ | | | |
| Propagation | [17] | | ✓ | | |
| Publishers, news, users relations | [18] | | ✓ | | |
| User features | [19] | | ✓ | | |
| News-topics inconsistency graphs | [20] | ✓ | ✓ | | |
| Content + social events integration | [21] | ✓ | ✓ | | |
| Triplet search in KGs | [22] | | | ✓ | |
| Rules extracted from fact patterns | [23] | | | ✓ | |
| Similarity with search engines results | [24] | | | | ✓ |
| Claim-evidence embeddings | [25] | | | | ✓ |

analyze the differences in terms of coherence and structure between deceptive and truthful narratives [14] have been used as features to detect misleading content. Methods such as word embeddings [26] have also been studied, which, when extended to the entire document, provide their ability to analyze the context in which a word appears, in order to identify typical patterns of disinformation and sensationalist language. It has even been found that the correct generation of these embeddings by training with the appropriate data [16], allows obtaining with simple classification models, results similar to or superior to those obtained with more complex models [27]. Another strategy used to detect disinformation based on the content of the document is the currently ubiquitous Transformer models [28]. On the one hand, taking advantage of its capabilities to extract latent features from the text without the need for elaborate feature design, and apply using transfer leaning the information obtained during its pre-training together with a fine adjustment in the target datasets to this detection task [15, 29]. On the other hand, Transformers can also be used to generate contextual embeddings that capture the meaning of words, as well as subtle cues that can characterize misleading information. In these cases, transformers are typically used as components of an ensemble model [30] that contains other components dedicated to the processing of intermediate information and the final classification of documents. These methods can provide good results although those based on writing styles, text analysis, and discourse analysis, require a careful design of features, and the results of the Transformer models can be very difficult to explain. However, the main disadvantage of all these methods is that they could be circumvented by an agent who can mimic the style of legitimate news even by incorporating misleading content. It is also worth mentioning in this group the existence of multimodal approaches that include other types of resources such as the images contained in the article [31, 32].

*Document context based.* The other strategy of detecting fake news is the use of context information that can be extracted from the document to be evaluated. One of the basic features is the speed of propagation [33] since this type of news tends to spread faster and more widely than real news. The implementation of this idea must be carried out in an environment where this type of information is available. For example, by consulting the Twitter API we can generate a propagation tree and a stance network from which various network features can be aggregated to determine the truth value of a tweet [17]. Other methods are based on the

relationships that exist between publishers, news, and users, given that publishers with some bias and low-reputational users are often associated with misleading content [18]. Features extracted from the user profile can also be valuable to detect this type of content [19]. The main disadvantage of these approaches is that they depend entirely on the existence of such contextual information so that if it is not available or not enough time has elapsed for it to be generated, it will be unable to assess the veracity of a news item.

*Mixed strategies*. More recently, mixed strategies have also been developed where both textual and context information are integrated. Among them we can highlight the creation of inconsistency graphs from the stance relationships between articles and topics, to which an energy flow model is applied to identify the most important nodes and determine their veracity according to the reliability of the sources of these nodes and their counterparts [20]. Another option in this sense is to take advantage of the power provided by Transformer models [21] to integrate both the content of the article and the social context of it (likes, shares, replies, etc.). While these mixed strategies should undoubtedly improve performance in terms of relying solely on content or context, they also inherit the same weaknesses described above.

**2.1.2 Claim based fake news detection.** This strategy is based on extracting the claims contained in the article (we could also consider it based on the content), identifying the most relevant ones (check-worthiness) and determining their veracity (fact-checking), assuming that some false claim would make the entire article false. For this reason, determining the veracity of an article it is usually broken down into these two tasks.

*Check-worthiness*. To extract claims from an article the most natural method is to break it down into sentences since these are the smallest syntactic constituent capable of expressing a statement. This task is relatively straightforward and can be performed by multiple NLP frameworks. Once the article is divided into sentences, we must determine which of them need to be verified. To achieve this, different types of features can be extracted such as *TF-IDF* vectors, sentiment scores, word and part-of-speech counts, entity types [34], topics modeled with *LDA*, entity stories [35], sentence embeddings, sentence positions in discourse segments [36], word embeddings, syntactic dependencies [37], and machine learning methods such as Multinomial Naive Bayes classifiers, Support Vector Machines, Random Forest classifiers, Multilayer Perceptrons and Recurrent Neural Networks. The emergence of Transformer models [28] and transfer learning has changed the rules of the game in the area of natural language processing, and it has also done so in this specific task. From that moment on, the systems that have obtained the best performance in different labs [38] have been those based on Transformer models such as BERT [39] or RoBERTa [40]. Training these models with large corpora in an unsupervised way and making a fine adjustment with the target dataset, allows us to extract latent features using only as input the sequence of tokens that form the phrase to be analyzed, thus avoiding the design of elaborated features.

*Fact-checking*. Once we have identified the factual and relevant sentences (claims) the next step is to determine the veracity of them. Here we find two main approaches, on the one hand we can transform the sentences into triplets (subject, predicate, object) and somehow verify these triplets against an existing knowledge base, and on the other hand we can try to verify the claim using some measure of similarity with respect to given evidence.

The first case can be raised as a search problem in a knowledge graph by making the truth value of a statement related to the distance in the graph from its subject and object entities [22], considering that crossing very generic entities increases the distance from other much more specific entities. A fact is defined as a triplet that has the form (subject, predicate, object). Assuming we are dealing with factual and relevant sentences that are not too complex, these sentences will contain a single claim that can be expressed in the form of a subject-predicate-object triplet (SPO triplet). The most direct way to analyze the veracity of a sentence of this

type is to collate that SPO triplet against an existing knowledge base (KB), where facts are stored as a directed graph in which subject and object are nodes connected by an edge representing the predicate. For this, it is necessary to divide and assign the different tokens of the sentence to each of these three entities: subject, predicate, and object. Another method based on graphs is the search for patterns during training that fit the facts in order to extract from them a series of rules that allow the evaluation of previously unobserved triples [23]. The main limitation of these methods based on knowledge graphs is the incompleteness of the information contained therein that leads to many claims cannot be verified.

Alternatively to the use of graphs, the possibility of using unstructured information to validate facts has also been explored. We can perform a search on Google and Bing for each claim to verify, retrieving snippets and web pages from reliable sources and calculating the similarity between the claim and the information retrieved [24]. Another approach is to build word co-occurrence graphs for claim and evidence, encode the semantic dependencies using graph gated neural networks, and finally incorporate the claim and evidence embeddings along with speaker and publisher embeddings into an attention mechanism that feeds a final layer of prediction [25]. With these systems based on searches for evidence and similarity we avoid the problem of incompleteness but rely on the reliability of the sources we consult to obtain such evidence.

**2.1.3 Fake news detection challenges.**   In general, none of the mentioned strategies is completely satisfactory due to the limitations indicated above, so we have a lot of room for investigating new methods of detecting fake news.

In addition, the use of neural networks in general and more powerful models such as Transformers have as a counterpart the lack of explainability since they basically behave like black boxes. Explainability is desirable in any machine learning task and the detection of fake news is no exception, so efforts are underway in this area. For example, [41] have analyzed three existing fake news detection models with three explainers: *Captum* [42], *SHAP* [43] and LIME [44], confirming that it is necessary to develop more interpretable explainers to increase user confidence in these systems. [45] perform a similar analysis using *LIME* and Anchor [46] on two fake news detection models based on *LSTM* and *BERT* respectively. A perhaps more desirable alternative to these external explainers is the one used by [47] incorporating into the model its own form of explanation based on coding the content of the news item along with the comments of the users and linking both through a mechanism of co-attention. Other approaches [48] manage explainability by trying to imitate the way in which humans evaluate the veracity of news, building a claim-evidence graph from which they reason using an adapted kernel graph attention network.

The system that we propose in this paper can be considered a claim-based system which also uses latent features, although differs from the previous ones in several aspects: 1) we carry the burden of detecting false news at the sentence level but we maintain as objective the evaluation of the news item, 2) we in turn break down sentences into subject, predicate, and object by exploring how Transformer models can use this information to evaluate the truthfulness of sentences, 3) we enrich the information of these sentences by means of terms proper to the domain of health (UMLS), and 4) we propose an ensemble model that allows integrating the results of the two previous points.

## 2.2 Fake news datasets

Since it is a fundamental element for training machine learning models in general and specifically for those used in the detection of fake news, researchers in this field have developed in recent years various datasets oriented to this task, often associated with a shared task or conference lab.

If we analyze these datasets, we can find certain characteristics that differentiate them from each other and make them more or less suitable as a tool for training and evaluating fake news detection models. The first is the way the truth value has been annotated. Many of them [49] rely on existing verification websites (e.g., Politifact, Snopes, Health Feedback) from which they retrieve this class information and usually the text of the news, accessing the links available on these sites. The disadvantage of this method is that we are limited to news already evaluated by these entities. In other cases fake news are fabricated altering real news manually [50, 51], which can lead to texts that do not accurately reflect the style of a "true" fake news. Another option is manual annotation [52], which requires having experts in that field, or relying indirectly on the verification entities mentioned above. We can also use as a reference the credibility of the source that publishes the news [36]. This allows to increase the scalability of the corpus by being able to easily incorporate new articles that are automatically annotated, but for certain media that are in an intermediate zone of credibility, these annotations would not be really reliable.

Another characteristic to consider in these datasets is the type of instance used. We can find from the full article [53], passing through posts on social networks or tweets [54], to reach smaller units such as sentences or claims [51]. These instances, especially the first ones, can be enriched by adding a social and temporal dimension, for example by searching the Twitter API with keywords [55] contained in the article. Other proposals are more oriented towards journalistic sphere [56] and break down the annotation of a news in its formal components (headline, subtitle, lead, body, conclusion) and the questions to answer (who, how, where, what, why, who). Depending on the type of news source we want to analyze, one type of dataset or another may be more interesting.

Regarding annotation, depending on the granularity of the evaluation we need and the features we want to use, we can consider aspects such as using binary [55] or multiclass [49] labels, or incorporating additional information such as the speaker, context or justification [57].

A final aspect to keep in mind is the domain on which the news included in the corpus deals with. This can be general [55] or specific to some collective or theme, such as health [54] or more specifically to the COVID [53]. Here the suitability of one type of dataset or another depends on our objective domain.

In this work we have collected a dataset focused on detecting fake news in the field of health where, unlike those mentioned above, each sentence present in a news item has been manually annotated both in its check-worthiness and in its truth value. This allows a fake news detection system to access an additional level of detail to determine the veracity of a news item, which can be decisive in improving their behavior. On the other hand, by using sentence as a minimum unit, we can incorporate instances such as posts on social networks that do not have a formal journalistic news structure.

Finally, it is worth mentioning among the efforts that are being carried out to fight against disinformation, initiatives such as *CheckThat! Lab* [58] included in the *CLEF* (Conference and Labs of the Evaluation Forum). In this laboratory, in which we have participated in the last editions, three types of tasks are carried out: determination of the check-worthiness of tweets and political debates sentences, claim retrieval from previous fact-checked claims, and fake news detection of complete news articles, allowing teams from different countries to verify and compare their proposals with the rest of the participants.

## 3 Definitions and formulation of the problem

In this paper we use the term news item or article to refer to a text obtained from an online newspaper or a blog, or a post on a social network.

Given a dataset of news items $N = \{n_1, \cdots, n_m\}$, each of these news items can be represented as a tuple $n_i = (t_{Ni}, c_i)$ where $t_{Ni} \in T_N$ is the textual content of the news item $i$, and $c_i \in C$, $C = \{F, T\}$, is its class label associated with the truth level of the news item.

The textual content $t_{Ni}$ of the news item $i$ can in turn be divided into sentences forming a set $S_i = \{s_{i1}, \cdots, s_{in}\}$. Each of these sentences is formed by the triplet $s_{ij} = (t_{Sij}, c_{CWij}, c_{FCij})$ where $t_{Sij} \in T_S$ is the textual content of the sentence $j$, $c_{CWij} \in C_{CW}$, $C_{CW} = \{NA, NF, FNR, FRC, FR\}$ is the class label indicating the check-worthiness assessment of the sentence, and $C_{FC} = \{F, T\}$, is the class label associated with the veracity of the sentence.

For each factual and relevant sentence $s_{ij} \in S_i$ we can extract a triplet $(sub_{ij}, pred_{ij}, obj_{ij})$ called triplet SPO in which $sub_{ij}$ corresponds to the subject, $pred_{ij}$ to the predicate or main verb, and $obj_{ij}$ to the object of the sentence $j$ included in news item $i$. We will represent the set of all SPO triplets of news item $i$ as $SPO_i$.

## 3.1 Problem formulation

Given the dataset $N$, we define the problem of detecting fake news as the determination for each news item $n_i \in N$ of its class value (veracity) $c_i \in C$ based on the information of the $S_i$ sentences it contains. Formally $f_{FN}: S \to C$ such that,

$$f_{FN}(x) = \begin{cases} F, & \text{news item } x \text{ has at least one false sentence.} \\ T, & \text{otherwise.} \end{cases}$$

To do this, first of all, it is necessary to select the $S_{FRi} \subseteq S_i$ sentences that are worth checking, thus defining the check-worthiness classification function $f_{CW}: T_N \to C_{CW}$ as,

$$f_{CW}(x) = \begin{cases} NF, & \text{sentence } x \text{ does not contain any factual statement.} \\ FNR, & \text{sentence } x \text{ does not contain any factual statement worth checking.} \\ FRC, & \text{sentence } x \text{ contains more than one factual statement that is worth} \\ & \text{checking or is very complex.} \\ FR, & \text{sentence } x \text{ contains a factual statement that deserves to be verified.} \\ NA, & \text{otherwise.} \end{cases}$$

Finally, to determine the veracity of the factual and relevant sentences $S_{FRi} \subseteq S_i$ we define two classification functions: $f_{SPOFC}: SPO \to C_{FC}$, and $f_{TextFC}: T_S \to C_{FC}$. In the first one the input is formed by the SPO triplets extracted from these sentences, while in the second function the input is the text of the sentence. The common definition for both functions would be:

$$f_{FC}(x) = \begin{cases} F, & \text{sentence } x \text{ is false.} \\ T, & \text{sentence } x \text{ is true.} \end{cases}$$

## 4 Objectives and contributions

In this section we state the main research objectives and research questions that derive from it, as well as the contributions made in this work.

## 4.1 Main research objectives

RO1) Improve the results of pre-trained models in sentence fact-checking including syntactic and medical domain-specific information.

RO2) Develop a system based on language models that are precise enough to autonomously assess the veracity of news in the field of health. This system must, in turn, allow to capture news evaluated by fact-checking entities and related to the medical domain, as well as the annotation of their sentences according to their check-worthiness and their truth value. As a by-product of this objective, we want to obtain a corpus of medical news where we have the truth value information at the sentence level, thus providing information on the reasons for the truth value assigned to each news item.

## 4.2 Research questions

RQ1) Is it possible to use the implicit knowledge provided by language models (LM) to verify the veracity of sentences without resorting to explicit knowledge bases?

RQ2) Can the results of these models be improved by introducing additional information such as the syntactic structure of the sentence and medical concepts?

RQ3) Is it possible to use a LM-based system and provide interpretable results, which also identify the points of the document that make it false?

## 4.3 Contributions

We have explored how to incorporate additional information to the knowledge available in the language models to try to improve the results of fake news detection in the medical domain. We have also explored different ways to take advantage of this information implicit in language models in a more efficient way. As a result, the main contributions of the work can be summarized as follows:

- Development of **new fact-checking classification models** of sentences based on Transformers that **incorporate syntactic information** in their input in the form of SPO triplets. This has allowed us to use more efficiently the implicit knowledge accumulated in a Transformer model during the pre-training phase making a fine adjustment with our corpus with the SPO triplets extracted from the sentences. In this way we were able to take advantage of this type of information without resorting to the use of external knowledge bases such as Wikidata or Yago.

- To complement the information structured in the form of **SPO triplets** with the **concepts of the medical domain** extracted from the sentences (CUIs), we have **developed an ensemble of Transformer and feed forward neural networks** that has allowed us to handle both features simultaneously in the process of classifying sentences according to their truthfulness. To do this, we have chosen to use the UMLS terminology and extract these medical concepts from the text of the sentences, associating each token of a sentence with its unique identifier (CUI), if it exists at all. The descriptions of these CUIs are also used in the classification process.

- We have carried out an **in-depth evaluation** of the different alternatives considered, which has allowed us to determine the improvements to which the different sources of information under consideration could lead. We have also explored the use of Transformer models that are currently the state of the art in the task of fact-checking sentences. The results show that, when we **combine SPO triplets** (RQ1) formed by the description of the CUIs present in the sentence (RQ2) with the **text of the sentence**, the result exceeds (F1: 0.698) that obtained by the Transformer models fed with unstructured text sequences (F1: 0.686) (RO1).

- In the classification of fake news, we have evaluated the feasibility of **using sentences to determine the truth value of complete news items**. To do this, we have applied a simple rule-based heuristic to use the information that comes to us from the sentences. Although this strategy is not capable of surpassing the results obtained with a Transformer model fed with the first $n$ tokens of the news (F1: 0.747), it helps us to **identify a large proportion of suspected fake articles** (F1 = 0.703). In addition, it provides an **additional level of explanation** (RQ3) by given the check-worthiness and fact-checking annotations at the sentence level.

- We have developed a **novel user-friendly methodology for collecting and annotating fake news**. This has allowed us to create a system that automatically crawls various internet sites to collect medical domain news and also include information on their veracity. Our system follows a sequence of processes to annotate sentences in the news based on their check-worthiness and truth value, and finally classifies the news items (RO2). To our knowledge, there is currently no system specifically designed to collect, facilitate annotation, and classify health-related news in an integrated manner.

- Thanks to the methodology introduced above, we have built a **new corpus of articles from the medical domain** composed of 317 articles and a total of 10335 sentences that are **annotated both in their check-worthiness and their truth value**. This level of detail provided by sentence-level annotation distinguishes it from other published corpora where annotation is done only at the article level. This collection of annotated news has been used to evaluate the different models mentioned above.

- The developed **innovative system** can also **operate autonomously**. In this mode, the predictions of the machine learning system are accepted automatically for the different phases of the process.

## 5 Methods

As mentioned in previous sections, in this work we have developed a pipeline composed of different stages that allows us to automatically collect news items related to the health topic from fact-checking sites, evaluate the sentences included in these documents according to its check-worthiness and truth value, and assign a final classification for every item to incorporate it into the dataset. All this is done from two perspectives: as an automatic classification tool that allows to determine the degree of veracity of a news item in an unattended way, and as an assistant for an annotator that confirms at each step the results of the different classification models used, i.e., the system can operate either attended or autonomous.

Fig 2 schematically represents the developed pipeline and the grated boxes indicate the validations that the annotator must perform. If the news item has been marked to skip validations, only the initial validation "Item Selection" would require intervention. As we can see, the pipeline is composed of the following stages, assuming the system is in attended mode:

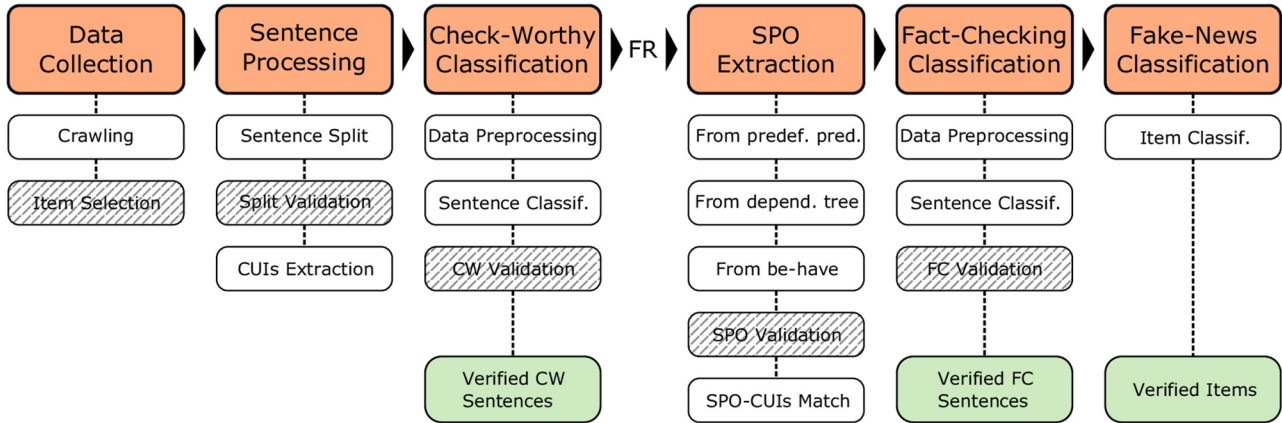

**Fig 2. Fake news detection pipeline.** In this detailed process diagram, shaded ones indicate a user intervention to validate or correct automatically generated information.

1) Data collection: Using a periodic process, the fact-checking websites Snopes and Politifact are consulted every 24 hours, recovering new evaluated news items and trying to retrieve the full text of the original article. The user selects the appropriate news items for the dataset, debugs them, and discards the rest.

2) Sentence processing: The selected news items are automatically divided into sentences and the user validates that division, making subdivisions if necessary. With the resulting sentences, the Metamap application is used to extract the Concept Unique Identifiers (CUIs).

3) Check-worthy classification: From a *training* dataset of manually annotated sentences, a classification of the new sentences is generated according to their check-worthiness. For this, a Transformer model is used to which fine-tunning has been done with this *training* dataset. The result is validated by the user.

4) SPO extraction: On the sentences classified as factual and relevant (FR) in the previous stage, the subject-predicate-object (SPO) triplet of each of them will be extracted, from predefined predicates, from the syntactic tree and dependencies generated by the Stanza tool, from predicates built with the be-have verbs and if none of the previous searches has obtained the SPO triplet, it is annotated manually. Subsequently, the tokens of the triplet are aligned with the CUIS extracted previously.

5) Fact-checking classification: At this stage, an evaluation of the veracity of each sentence is carried out using a *training* dataset manually annotated. This annotation at the sentence level has been made based on the truth value assigned by the fact-checking website to the complete article, considering that not all the sentences have to be false in a fake article. The classification is carried out with a Transformer-FFNN ensemble model that can simultaneously receive as inputs, the SPO triplet extracted in the previous stage, the text of the sentence, the CUIs associated with each token of the sentence, or the description of these CUIs.

6) Fake news classification: In this last stage, each news item is classified according to the fact-checking classification of its factual relevant sentences. This final classification is made with a simple rule, that is, if there is any false sentence the news item is false, otherwise, the news item is true.

## 5.1 Data collection and annotation process

The methodology presented in this paper has allowed us to gather a collection of news items and manually annotate them, to compile an evaluation corpus with which to develop and analyze the performance of our systems. This section describes how we collected the news items, the annotation criteria used, and some statistical data from the corpus. The dataset is publicly available and the access link can be found in Section 9. It is composed of the URLs that allow access to the original news items, and for each of them, the sentence-level annotation of both: fact-checking and check-worthiness. Its structure and statistics are described later in Section 5.1.4.

**5.1.1 Data collection and selection of news items.** The collection has been initiated with the news items from the Health Feedback website. Then it has been augmented with more up-to-date data from two of the main fact-checking sites. To do this, we have developed a process that checks if there are new entries in the fact-checking websites *Snopes* and *Politifact*. On the first of these sites the list of news items of the "medical", "health" and "health-politics" categories are checked. With each news item retrieved, the main claim, the review summary, the URL of the original news item and the evaluation made by the fact-checking website are located, converting this evaluation to uniform class values: *T* = (*true*, *correct attribution*, *mostly true*), and the rest of the values are assigned to *F*. In *Politifact* the new item in the categories "health-check" and "coronavirus" are checked, and as with *Snopes*, we try to retrieve the review summary, the URL of the original news item and the evaluation, with the following correspondence of class values: *T* = (*true*, *mostly true*), and *F* = (*barely true*, *half true*, *pants fire*, *false*). This process runs every 24 hours and the retrieved items are stored. In this way the news items are accumulated until they are validated.

We have organized the workflow into several revision levels that determine the stage a news item is at. Newly captured items are at the lowest revision level. In the review made by the annotator, different aspects are checked before the item is considered valid, which for example would be discarded in the following cases:

- Content is a video or image and there is no textual information (*Does Video Show Athletes Fainting Due to COVID-19 Vaccine?*).

- The news item is not related to health, is only tangentially related, or not relevant enough (*Is Budweiser Giving Away a Free Beer to People Who Have Been Vaccinated?*).

- The original URL could not be located even manually.

Items that are not discarded due to the above criteria are passed to the next level of review.

To manage this information, we have developed a web application that allows us to edit the incoming items to add missing information, correct poorly captured data, change the review level as well as register items that have not been referenced by the two fact-checking entities used. If we register an item of which we do not have an evaluation by a fact-checking website we can indicate that this item will be treated as an instance to be evaluated and, in this case, it will not stop at the stages of manual validation, following the workflow until obtaining an evaluation of its level of veracity.

**5.1.2 Check-worthy annotation.** Once the news items to be considered have been selected and the sentences present in each news item have been obtained through the process that will be described in Section 5.2.1, the next step is to determine which ones are worth checking. Sentences of this type must be factual (they must incorporate a statement about some fact) and relevant (the fact must have sufficient interest to be verified). To do this, we have selected class values similar to those used by other authors [34]: *non-factual*, *factual non relevant* and *factual*

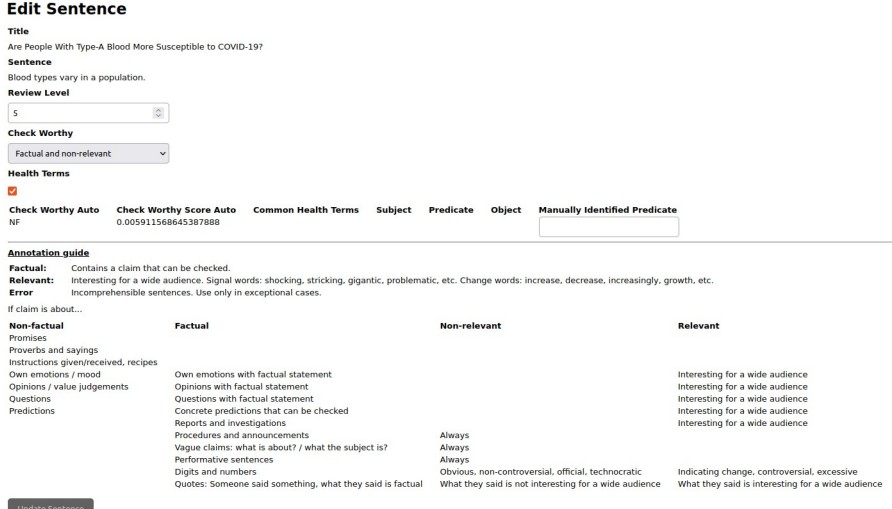

**Fig 3. Sentence check-worthy annotation screen.** At the top we can make the check-worthy annotation, manually indicate the predicate, and modify the revision level to send the sentence to the next stage. At the bottom we have the check-worthy annotation guide.

*relevant*, also adding a value for sentences that do not make sense to evaluate (*not applicable*) and another for factual relevant sentences that are too complex to evaluate their veracity automatically (*factual-relevant-but-complex*).

To make the annotation of the sentences we have followed some criteria (detailed in Section 9) derived from those used by [59]. First, we evaluate whether the sentence refers to a fact that can be proven and evaluated as true or false. If so, the sentence will be factual and otherwise it will be non-factual. Then, for the sentences considered factual, we have determined whether they are relevant or not.

With these criteria and using as an aid the screen shown in Fig 3, the annotation of 100 sentences has been made by the three authors obtaining a *Fleiss' kappa* inter annotator agreement [60] of 0.635. After verifying that the criteria used has led us to a substantial agreement, we have annotated the approximately 10,000 sentences resulting from processing the 327 news items that the corpus currently has.

**5.1.3 Fact-checking annotation.** At this essential stage in the process of determining the veracity of a news item, only sentences that have been previously annotated as factual and relevant arrive.

As in the previous stage, a manual annotation of the approximately one thousand sentences of this type has been made. To do this, in each news item we have reviewed the evaluation of the article made by the verification entity. For example, *Snopes* usually identifies suspicious claims within the article and their level of truthfulness. Based on this, we have searched for the sentences where these claims appear to label them as false. We have considered the rest of the sentences true. In specific cases where false claims were not clearly identified, we have turned to reliable sources such as Wikipedia or MedlinePlus to verify suspicious sentences.

**5.1.4 Compiled dataset structure and statistics.** The result of these collection, selection, and annotation stages, together with the processes applied to these data that will be described in Section 5.2, has been a set of sentences grouped into news items, annotated in both their check-worthiness and their fact-checking. The final dataset consists of 10335 sentences divided into 1132 *factual relevant* sentences whose fact-checking values has been annotated, and 9203

**Table 2. KEANE dataset statistics.**

| Check-Worthy Class | Fact-Checking Class | Sentence Count | | Fake News Class | News Item Count | |
|---|---|---|---|---|---|---|
| Not Applicable | Not Applicable | 26 | (0.25%) | | | |
| Not Factual | Not Applicable | 3735 | (36.14%) | | | |
| Factual Not Relevant | Not Applicable | 5093 | (49.28%) | | | |
| Factual Relevant but Complex | Not Applicable | 349 | (3.38%) | | | |
| | | | | Not Applicable | 53 | (16.21%) |
| Factual Relevant | False | 674 | (6.53%) | False | 181 | (55.35%) |
| Factual Relevant | True | 458 | (4.43%) | True | 93 | (28.44%) |
| **Total** | | **10335** | | | **327** | |

*not factual, factual not relevant, not applicable,* or *factual relevant* sentences that contain more than one claim and are therefore too complex to be analyzed. Detailed statistics of the number of sentences in each class are shown in Table 2. In this table we can also see the number of news items of each fake news class. Although all news items have an annotation from the entities used in their collection, news items with all their sentences with fact-checking class *not applicable* and news items that do not have any sentences with check-worthy class *factual relevant* have been considered *not applicable.* The main topics present in the dataset are shown in Table 3. As can be seen, a significant percentage is related to COVID or associated topics such as face masks. This is because part of the collection work took place during the pandemic. Nevertheless, in the table we can observe a high variability of topics.

Table 4 shows the information included for each instance of the dataset.

## 5.2 Data processing and classification

This section details the different information extraction and classification processes that have been developed to maintain the workflow in the pipeline once we have selected the news items to be validated and/or incorporated into the corpus. As mentioned above, we can mark it so that the system works in an attended manner and waits for user feedback at certain times, or we can let the system process it in an unattended way until we obtain an estimate of its truth value. These processes have also been used to perform the initial compilation of news items.

**Table 3. KEANE topics distribution.**

| Topic | News Item Count | Percentage |
|---|---|---|
| Cancer | 22 | 6,73% |
| Cannabis | 8 | 2,45% |
| COVID | 80 | 24,46% |
| Face masks | 11 | 3,36% |
| Flu | 13 | 3,98% |
| Health advices | 9 | 2,75% |
| Heart diseases | 7 | 2,14% |
| Mental diseases | 18 | 5,50% |
| Nutrition/obesity | 39 | 11,93% |
| Other medical studies/news | 93 | 28,44% |
| Smoking | 7 | 2,14% |
| Non-COVID vaccines, anti-vaccines | 20 | 6,12% |

**Table 4. KEANE dataset instance structure.**

| id | Unique identifier of the sentence. |
| --- | --- |
| item_id | Identifier of the article to which the sentence belongs. |
| item_url | URL from which the article can be downloaded. |
| sentence_id | Determines the order of the sentence within the article. When it is zero, it is the title of the article. |
| check_worthy | Check-Worthy label of the sentence that can take the values *NF*, *FR*, *FNR*, *NA*, *FRC* ("Not Factual", "Factual and Relevant", "Factual and Non-Relevant", "Not Applicable", "Factual, Relevant but Complex" respectively). |
| sentence_class | Truthfulness label of the sentence that can take the values *F*, *T* ("False", "True" respectively). |
| instance_type | Dataset to which the sentence belongs. It can take the values *Train*, *Test*. |

**5.2.1 Sentence processing.** At this stage, the text of the news items that have been selected is cleaned by replacing tabs and line breaks with spaces, eliminating multiple consecutive spaces and adding periods at the end of certain sentences that end in double quotation marks. The article is automatically divided into sentences and this division is validated by the annotator. The resulting sentences are captured by the process detailed below.

Since we have focused this framework on the detection of fake news in the field of health, we consider that the use of standardized concepts associated with words that appear in this context can be a type of useful feature in order to improve the performance of the models developed.

Specialized terminologies such as the Unified Medical Language System (UMLS) [61] are very useful for characterizing documents in the medical domain. UMLS is a repository of biomedical vocabularies and a set of tools that allows, among other things, extract concepts, relationships, or knowledge from text. It is composed of three sources of knowledge: the *Metathesaurus*, a large multilingual vocabulary containing information on biomedical and health-related concepts, different names for these concepts, and relationships between them; the *Semantic Network* that provides a consistent categorization (semantic types) of all the concepts represented in the UMLS Metathesaurus and provides a set of useful relationships between these concepts; and the *SPECIALIST Lexicon* which provides lexical information on many medical and common English terms. Each concept or meaning in the Metathesaurus is associated with a single, permanent concept identifier (CUI) that can also be grouped into one or more semantic types. For example, the term *diabetes* has the CUI *C0011847* and belongs to the semantic type *T047 Disease or Syndrome*.

A simple way to access the resources provided by UMLS is to use tools such as cTakes or MetaMap. These tools based on natural language processing and computational-linguistic techniques allow to discover concepts of this Metathesaurus in texts by assigning CUIs to the identified tokens.

We have chosen to use this terminology to associate CUIs to sentence words, and as a tool to extract these medical concepts from the text of the sentences, we have selected the MetaMap application since it is easy to manage through batch processes. To do this mapping, for each sentence, we apply a "Part-of-Speech Tagger", a "Word Sense Disambiguation", and *MetaMap* itself to generate the sentence analysis and extract its concepts. We link the discovered CUIs with the tokens of the sentence, assigning a tag [UNK] when there is no CUI associated with a token. Since *MetaMap* can generate several candidate CUIs for a given concept, we select the one with the highest score assigned by this tool.

In addition, to restrict the type of extracted concepts to those that have more to do with diseases and treatments, we only map the CUIs that belong to 127 semantic types included within the semantic groups listed in Section 9.

**Table 5. CheckThat! 2021 English check-worthiness estimation in tweets results sorted by MAP.**

| Rank | Team | MAP | MMR | RP | P@1 | P@3 | P@5 | P@10 | P@20 | P@30 |
|------|------|-----|-----|----|-----|-----|-----|------|------|------|
| 1 | **NLP&IR@UNED** [68] | **0.224** | 1.000 | 0.211 | 1.000 | 0.667 | 0.400 | 0.300 | 0.200 | 0.160 |
| 2 | Fight for 4230 [69] | 0.195 | 0.333 | 0.263 | 0.000 | 0.333 | 0.400 | 0.400 | 0.250 | 0.160 |
| 3 | UPV [70] | 0.149 | 1.000 | 0.105 | 1.000 | 0.333 | 0.200 | 0.200 | 0.100 | 0.120 |
| 4 | bigIR | 0.136 | 0.500 | 0.105 | 0.000 | 0.333 | 0.200 | 0.100 | 0.100 | 0.120 |
| . . . | . . . | . . . | . . . | . . . | . . . | . . . | . . . | . . . | . . . | . . . |
| 10 | TOBB ETU [71] | 0.081 | 0.077 | 0.053 | 0.000 | 0.000 | 0.000 | 0.000 | 0.050 | 0.080 |

All of these processes, except for validating the division of text into sentences, are executed unattended, making the resulting sentences the input of the next stage.

**5.2.2 Check-worthy classification.** According to our experience after participating in task 1A *"Check-Worthiness Estimation of Tweets"* of CheckThat! Lab 2021 [62], where we reached the first position (Table 5) in the English version of the task, a Transformer model that uses as input the sequence of tokens that make up a sentence or tweet is able to extract the latent features that determine if it is worth checking. That is why we have developed a classifier of this type and have evaluated with the pre-trained models *emilyalsentzer/Bio_ClinicalBERT* [63], *dmis-lab/biobert-v1.1* [64], *bionlp/bluebert_pubmed_uncased_L-24_H-1024_A-16* [65], *funnel-Transformer/intermediate* [66], *bert-base-cased*, *bert-base-uncased* [39] (which from now on we will identify them respectively as *ClinicalBERT*, *BioBERT*, *BlueBERT*, *Funnel*, *BERT Cased*, *BERT Uncased* and *ALBERT* for short), and *albert-base-v2* [67], finally selecting the *BERT Uncased* model for both binary classification and multi-class classification.

To fine-tune the model we randomly divided the overall dataset of 10335 sentences in the *training* and *test* datasets with a 0.8:0.2 ratio, and divided the *training* dataset again with the same ratio to extract the *dev/validation* dataset. In this way we have obtained 6708 sentences in the *training* dataset, 1678 sentences in the *dev/validation* dataset and 1949 sentences in the *test* dataset. The distribution of instances by class is maintained in the resulting datasets (*training +dev/validation*: FNR = 49.17%, FR = 10.67%, FRC = 3.55%, NA = 0.27%, NF = 36.33%; *test*: FNR = 49.77%, FR = 12.16%, FRC = 2.62%, NA = 0.15%, NF = 35.30%, where FNR: Factual Not Relevant, FR: Factual Relevant, FRC: Factual Relevant but Complex, NA: Not Applicable, NF: Not Factual). This training is done during a variable number of epochs with a batch size of 16. The input sequence size has been set to 128 tokens, enough to hold most sentences.

This classifier is used to automatically annotate the sentences generated from new news items that arrive from previous stages. However, once processed these sentences remain waiting to be reviewed and validated manually before moving on to the next stage.

**5.2.3 Fact-checking classification.** With the manual annotation made on the factual and relevant sentences, a classifier has been trained to evaluate the veracity of the new sentences that are incorporated into the corpus.

To develop this classifier the main strategy we have followed is to use subject-predicate-object triplets (SPO) as a basic input data, but unlike in the usual approximations where knowledge graphs and distances between entities [22] are used to evaluate the veracity of a SPO triplet, we have developed a method based on the work by [72], in which subject, predicate and object are treated as sequences of tokens that are introduced to a Transformer model as input, separating the three sequences by a special token. The justification for this strategy is based on the fact that one of the tasks with which models such as *BERT* or *ALBERT* have been pre-trained is the prediction of the next sentence. Those sentences, being really contiguous segments of text, could be assimilated to subjects, predicates, and objects, so in a certain way,

the model can contain latent information from these SPO triplets. This allows us to incorporate external knowledge to a certain extent if we consider that the thousands of documents with which the pre-training has been carried out contain this type of factual information. On the other hand, this external knowledge can be updated as these models are pre-trained with more recent documents. Alternatively, we can also update this factual information if we perform fine tunning using recent documents. As far as we know, this is the first time this strategy has been applied in the fact-checking task.

Therefore, the first task that is performed at this stage is the extraction of subject, predicate and object in each factual relevant sentence through a series of incremental processes explained below. In all cases if one of the processes finds the SPO triplet in a sentence, the subsequent processes ignore it.

First of all, and in a similar way to what [73] does but with a different objective, we have selected manually a series of verbs in English that can represent relationships in the medical domain such as *cause*, *spread*, or *prevent* (a complete list is provided in Section 9), and we have created a process that generates more complex predicates by conjugating these verbs in their different verb tenses, adding negations, modals, etc. Using these extended predicates as a search pattern, we have examined each sentence and for each match we have considered as a subject all the tokens that appear in front of the matching segment, and predicated all the tokens that appear behind. If there are no matches, the process is repeated using as a search pattern predicates that only contain infinitives, present 3rd, present participles, and past participles.

The following process uses the Stanza tool [74] to parse the sentence. From the result obtained, we look for the dependency relationship *root* and the associated word is considered the predicate. The subject is the word that has the part-of-speech *NOUN* or *PROPN* and the dependency relationship *nsubj* (nominal subject) or *nsubj:pass* (passive nominal subject). If nothing is found, the part-of-speech *PRON* or *PRP* and the dependency relationship *nsubj* or *nsubj:pass* are checked. The object is the word that has the part-of-speech *NOUN* or *PROPN* and the dependency relationship *obj* or *obl*. Extended predicates are generated by querying the tokens that depend directly on *root* in the syntactic tree of the sentence and that also have part-of-speech *AUX*, *COP*, *PART*. The extended subject would be all the tokens before the extended predicate and the extended object all the tokens after the last token of the extended predicate. In this process we initially check if there is text in quotation marks in the sentence, in case it is a quote from someone who is affirming something. If so, what is outside the quotation marks is ignored. In case the analyzer has not been successful, it is also checked if there is a comma in the sentence separating two clauses. If so, the process is repeated for each clause considering also object the word that has the part-of-speech *ADJ* and the dependency relation *xcomp* (open clausal complement) or *ccomp* (clausal complement).

If we have not yet found predicate in the sentence, a third process looks for predicates with the verbal constructions *modal+be* (*can be*, *should not*, . . .), *be* (*is*, *are*, *won't have been*, . . .) or *have* (*has*, *will*, *have*, . . .), extracting subject and object from the predicate found as described above. With this we seek to capture more generic relationships of the "is-a" or "has-a" type, which can also be present in any sentence.

Once we have divided the sentence, we also generate a sequence of CUIs that can be used as an alternative input for the classifier through a process of aligning the tokens in subject, predicate and object with respect to the CUIs extracted in the sentence processing stage. When no CUI exists, it is replaced by the *[UNK]* token. We also generate another type of alternative input by substituting the CUI codes for their description in subject, predicate, and object.

If the predicate is not detected automatically, the user can indicate it manually so that that sentence continues processing. If he does not do so, the sentence will be discarded for the next stages.

The next step at this stage is the automatic classification of sentences. For this, different models of FFNN and Transformer have been developed and evaluated, using different types of input such as:

- The sequence of tokens of the complete sentence.

- The CUI codes extracted from the sentence.

- The sequence of tokens of subject, predicate, and object.

- The sequence of tokens of subject, predicate, and object, with spaces at the end of each of these elements. So that, for any sentence, predicate and object start at the same position in the sequence.

- SPO triplets containing the description of the extracted CUIs.

Table 6 shows an example sentence with the different input types extracted from it (the *@end@* label is used to show the end of the sentence in this example but is not actually added to the input).

To try to improve the performance of these classifiers, we have developed an ensemble model (Fig 4) that allows the simultaneous use of more than one of these inputs, thus taking advantage of the complementarity of the information they provide.

Different Transformer models have been trained with full sentences, and the three SPO triplet input types. We have also enabled the option to increase the vocabulary of the pre-trained model by adding words that appear in the *training* dataset but are not in that model. With this, we can incorporate medical terms that have recently appeared or were not in the documents with which the Transformers were pre-trained.

The *FFNN* classifier is composed of a hidden layer and an output layer, and has also been trained in the same *training* dataset with different hyperparameters such as the size of the hidden layer, the type of activation function, the type of input (*TF-IDF* vectors created with the CUIs present in the sentence or *TF-IDF* vectors created with the text of the sentence), the number of epochs and the option to use weights in the class values to compensate for their different distribution.

The result of runs with different hyperparameters in each of these two models is saved. In the ensemble model, we re-instantiate these models according to the hyperparameter configuration we want. These loaded models are executed within the ensemble in evaluation mode, that is, the value of their parameters is not altered by training. The first token of the last hidden layer of the Transformers (classification token) and the hidden layer of the FFNN are used as

**Table 6. Sentence example and the different types of input extracted.**

| Type | Example |
|---|---|
| Full sentence | Having a banana a day keeps the coronavirus away.@end@ |
| CUIs | [UNK] [UNK] C0939797 [UNK] C0439228 C0333118 [UNK] C0206419 [UNK] @end@ |
| SPO triplet variable length | Having a banana a day[SEP]keeps[SEP]the coronavirus away.@end@ |
| SPO triplet fixed length | Having a banana a day [SEP]keeps [SEP]the coronavirus away. @end@ |
| SPO triplet text CUIs | banana extract day[SEP]Retained[SEP]Genus: Coronavirus@end@ |

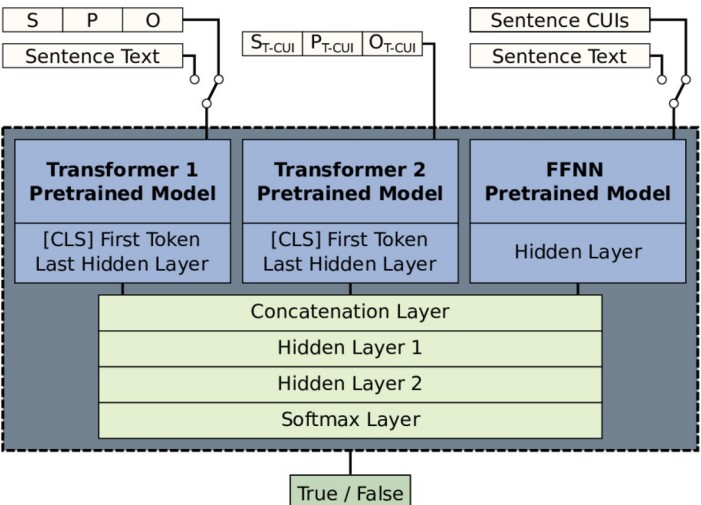

**Fig 4. Transformer-FFNN ensemble.** Of the five possible input types only three are used simultaneously. The top models are loaded in evaluation mode and only the parameters of the bottom layers are modified during training.

the input of the internal classifier formed by two hidden layers and an output layer. In Fig 4 we can see the types of input that can be configured in each block: full sentence, SPO triplets of variable length, and SPO triplets of fixed length in the first Transformer, SPO triplets with descriptions of CUIs in the second Transformer, and complete sentences or CUI codes in the FFNN.

Finally, after the execution of these models, we obtain a classification of the veracity of the factual relevant sentences that we can evaluate with respect to the annotation made manually.

**5.2.4 Fake news classification.** In this last stage we perform the classification of news items considering the fact-checking classification of their sentences. Since in this paper we want to emphasize the use of information at the sentence level, we have limited ourselves to using a simple heuristic: if there is a false sentence, the news item is false (F), otherwise, the news item is true (T).

To evaluate the results of this strategy compared to more traditional approaches, a classifier based on Transformer models that accepts as input the first *n* tokens of the news item has been implemented. This method has proven in various shared tasks [75] to obtain competitive results without the need to develop features adapted to the task of detecting fake news. Using a somewhat more elaborate approach, we achieved second place (Table 7) in Task 3 "Multi-class fake news detection of news articles" organized at CheckThat! Lab 2022 [76], using an

**Table 7. CheckThat! 2022 English fake news detection results sorted by macro-F1.**

| Rank | Team | True | False | Partially False | Other | Accuracy | Macro-F1 |
|------|------|------|-------|-----------------|-------|----------|----------|
| 1 | iCompass [77] | 0.383 | 0.721 | 0.173 | 0.080 | 0.547 | 0.339 |
| 2 | **NLP&IR@UNED** [78] | 0.446 | 0.729 | 0.097 | 0.057 | 0.541 | **0.332** |
| 3 | Awakened [79] | 0.328 | 0.744 | 0.185 | 0.035 | 0.531 | 0.323 |
| 4 | UNED | 0.346 | 0.725 | 0.191 | 0.000 | 0.544 | 0.315 |
| . . . | . . . | . . . | . . . | . . . | . . . | . . . | . . . |
| 25 | AI Rational | 0.296 | 0.000 | 0.196 | 0.090 | 0.098 | 0.117 |

ensemble model that combined a Transformer fed with the first 128 tokens of the article and a FFNN fed with LIWC features extracted from the article text.

Fig 5 shows an extract from an article external to the corpus on which the different stages of the pipeline have been applied until reaching the final veracity assessment.

**5.2.5 Pipeline training and inference times.** To conclude the Methods section, in this subsection we offer some data about the training and inference times that we have obtained in our tests.

As described above, we use three types of models: FFNNs, Transformers, and ensembles composed of the previous two. The training has been carried out using a workstation with a GPU, while the continuous execution of the pipeline in inference mode is carried out on a much more modest NUC-type PC equipped only with a mobile processor (Section 9 details the hardware used). During training we use the early stopping mechanism so the number of epochs is variable in each case.

In the check-worthy task, training a Transformer model such as *BERT Base Cased* has required almost 7 minutes in binary classification (6 epochs), and just over 16 minutes in multiclass classification (10 epochs). In the sentence fact-checking task we have verified that an FFNN model can be trained in less than 10 seconds (8 epochs), a BERT-type Transformer uses a little more than a minute (6 epochs), and the training of the ensemble model is performed in just under a minute (5 epochs). Therefore, training an complete ensemble model with two Transformers and one FFNN does not take more than 4 minutes. In the fake news detection task, the time needed to assess the veracity of an entire article based on its sentences is only a few seconds.

Regarding the inference times, the processing of a complete article with 96 sentences of which 5 are factually relevant needs a little over 12 minutes. A significant part of this time is consumed by the *MetaMap* process that extracts medical identifiers from the text of sentences.

# 6 Results

In this section we show the results obtained with the different models developed as part of the pipeline that allows to automatically annotate the collection of new news items, for the check-worthy, fact-checking, and fake news detection classification tasks. In all cases the evaluation measure used to determine the performance of each model has been F1 macro average, although accuracy, precision and recall have also been included.

## 6.1 Sentence check-worthy classification results

The first classification process in our pipeline aims to select the sentences that are worth checking. For this task, only Transformer models have been selected since this architecture provides high quality results without the need to perform complex feature design. A grid search has been carried out with the seven Transformer models mentioned in Section 5.2.2 and a maximum sequence size of 128 tokens. An early stopping mechanism has been implemented that stops training if the F1 measure has not been improved in the *dev* dataset in the last 3 epochs, recovering the saved model that had the best result for that configuration. The maximum number of epochs has been set to 10.

Furthermore, to alleviate the performance variability inherent to the Transformer models [80], each model has been executed for ten different random seeds and the average of the accuracy, precision, recall and F1 measures has been calculated to elaborate the results shown in Table 8 (binary classification) and Table 9 (multi-class classification). Next to each measurement, the standard deviation obtained on these ten random seeds is also shown.

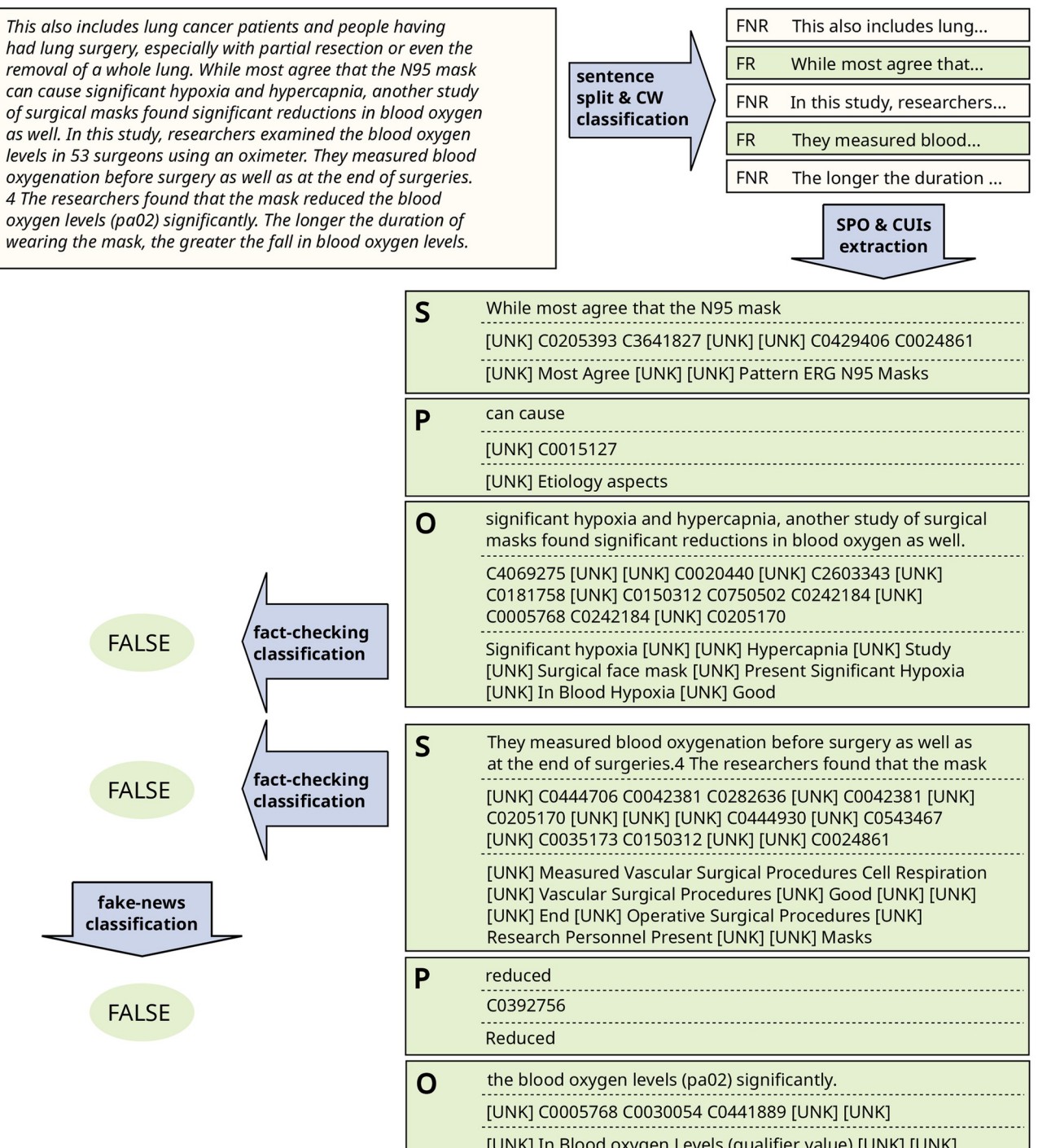

**Fig 5. Example of fake news detection (excerpt from an external article).** The article is divided into sentences and their check-worthiness is evaluated. From the factual and relevant sentences, the SPO triplets are extracted and the CUIs codes and their descriptions are aligned. The veracity of these sentences is evaluated, resulting in both being false. Finally, the veracity of the complete article is deduced from the veracity of the sentences.

**Table 8. Check-worthy binary classification results.**

| Classifier Type | Avg. Acc. | Avg. Prec.+ | Avg. Rec.+ | Avg. F1+ | Avg. Prec. | Avg. Rec. | Avg. F1 |
|---|---|---|---|---|---|---|---|
| BERT Uncased | 0.894±0.006 | 0.567±0.028 | 0.555±0.050 | **0.559±0.025** | 0.753±0.013 | 0.748±0.021 | **0.749±0.013** |
| BlueBERT | 0.900±0.003 | 0.609±0.023 | 0.508±0.064 | 0.551±0.035 | 0.771±0.008 | 0.731±0.027 | 0.747±0.018 |
| BERT Cased | 0.890±0.006 | 0.553±0.033 | 0.510±0.056 | 0.528±0.027 | 0.743±0.015 | 0.726±0.023 | 0.733±0.014 |
| BioBERT | 0.900±0.003 | 0.624±0.028 | 0.448±0.055 | 0.519±0.031 | 0.775±0.012 | 0.705±0.024 | 0.731±0.015 |
| Funnel | 0.900±0.009 | 0.551±0.196 | 0.493±0.178 | 0.519±0.183 | 0.742±0.107 | 0.725±0.081 | 0.731±0.093 |
| ClinicalBERT | 0.889±0.005 | 0.548±0.026 | 0.502±0.057 | 0.522±0.029 | 0.740±0.012 | 0.722±0.024 | 0.729±0.015 |
| ALBERT | 0.886±0.008 | 0.544±0.047 | 0.405±0.071 | 0.461±0.047 | 0.732±0.024 | 0.679±0.032 | 0.698±0.024 |

Results for different pre-trained models sorted by Avg. F1. In addition to accuracy, the table shows the values of precision, recall, and F1 measure for the positive class (label *True*), and using macro average.

**Table 9. Check-worthy multi-class classification results.**

| Classifier Type | Avg. Acc. | Avg. Prec.+ | Avg. Rec.+ | Avg. F1+ | Avg. Prec. | Avg. Rec. | Avg. F1 |
|---|---|---|---|---|---|---|---|
| BERT Cased | 0.712±0.008 | 0.547±0.022 | 0.544±0.060 | 0.543±0.030 | 0.659±0.010 | 0.582±0.013 | **0.611±0.010** |
| BioBERT | 0.715±0.007 | 0.607±0.032 | 0.480±0.054 | 0.533±0.024 | 0.659±0.011 | 0.566±0.011 | 0.602±0.008 |
| BlueBERT | 0.718±0.008 | 0.606±0.035 | 0.493±0.068 | 0.541±0.040 | 0.663±0.016 | 0.563±0.027 | 0.599±0.021 |
| ClinicalBERT | 0.700±0.009 | 0.590±0.028 | 0.463±0.071 | 0.514±0.041 | 0.656±0.010 | 0.563±0.019 | 0.596±0.014 |
| ALBERT | 0.704±0.013 | 0.557±0.026 | 0.467±0.067 | 0.505±0.043 | 0.644±0.014 | 0.565±0.020 | 0.593±0.017 |
| BERT Uncased | 0.706±0.008 | 0.576±0.026 | 0.468±0.068 | 0.513±0.039 | 0.637±0.027 | 0.557±0.020 | 0.586±0.019 |
| Funnel | 0.724±0.011 | 0.599±0.048 | 0.534±0.057 | **0.562±0.034** | 0.589±0.108 | 0.547±0.082 | 0.562±0.091 |

Results for different pre-trained models sorted by Avg. F1. In addition to accuracy, the table shows the values of precision, recall, and F1 measure for the positive class (label *Factual Relevant*), and using macro average.

As it can be seen, in Table 8 the *BERT Uncased* model obtains the highest F1 value in binary classification (0.749). If we now look at the values of F1 with respect to the positive class, we can see that in binary classification *BERT Uncased* continues to obtain the highest value (0.559). On the other hand, in the multi-class classification (Table 9), the highest value (0.611) is obtained with *BERT Cased*, and with *Funnel* for the positive class (0.562). In general, we see that the use of pre-trained models in the biomedical domain (*BlueBERT*, *BioBERT*, *Clinical-BERT*) does not provide a substantial improvement in this task. This indicates that the features that determine whether a sentence is worth checking do not depend heavily on the use of terms and language specific to this domain. Since we consider sentence check-worthiness estimation an auxiliary task within the pipeline, we have selected *BERT Uncased* as the default model.

## 6.2 Sentence fact-checking classification results

To evaluate the result in the fact-checking classification of sentences of the two mains strategies followed (the use of extracted CUI information, and the use of SPO triplets as input to Transformer models instead of traditional unstructured text sequences), we have carried out a grid search with different hyperparameters in the Transformer, FFNN, and ensemble models. We have considered as baselines the FFNN neural network whose input are TF-IDF vectors extracted from the text of the sentence and the different Transformer models also fed with the text of the sentence.

With the Transformer models we have used the pre-trained models *ClinicalBERT*, *Bio-BERT*, *BlueBERT*, *Funnel*, *BERT Cased*, *BERT Uncased* and *ALBERT*, with and without vocabulary expansion, the input types mentioned in 5.2.3 section, and a maximum of 10 epochs.

In the FFNN model we have used the activation functions *sigmoid*, *tanh* and *relu*, the hidden layer sizes 100, 500, 1000 and 1500, the input types CUIs or sentence text, with and without class weight compensation, and a maximum of 250 epochs.

For the ensemble model we have set the most promising hyperparameters of the previous two classifiers, Transformer and FFNN. From the Transformer we have selected the *BERT Cased*, *ALBERT*, *BioBERT* and *BlueBERT* pre-trained models, no vocabulary expansion, and a maximum of 10 epochs. The first four pre-trained models have been evaluated in the first Transformer component of the ensemble and *BlueBERT* has been configured in the second Transformer component that accepts SPO triplets formed by the descriptions of the CUIs. And from the FFNN we have used the *tanh* activation function, a hidden layer size of 100, without class weights compensation, and 250 epochs at most. In the three components of the ensemble, the random seed that had a higher F1 value was used. Based on this configuration, we have evaluated the different input types and hyperparameters: pre-trained model for the first Transformer component of the ensemble, one or two hidden layers, dropout of 0, 0.2 and 0.5, and a maximum of 10 epochs.

The evaluation has been carried out on a *training* dataset of 716 sentences, a *dev/validation* dataset of 179 sentences and a *test* dataset of 237 sentences. These datasets have been obtained based on the 1132 factual relevant sentences with the same partitioning strategy (0.8:0.2 + 0.8:0.2) used in the check-worthiness classification. The resulting distribution by classes has been maintained in each dataset (*training+dev/validation*: F = 59.55%, T = 40.45%; *test*: F = 59.49%, T = 40.51%). In addition, each hyperparameter configuration has been executed for ten different random seeds and their average has been calculated to elaborate the accuracy, precision, recall and F1 results, as was done in the check-worthiness classification. The same early-stopping mechanism has also been used.

Table 10 collect the results obtained on the *test* dataset for the accuracy, precision, recall and F1 measures. Table 11 shows the same information for the positive class (label *False*). As indicated above, the measurements shown are averaged for 10 different random seeds.

If we look at the results shown in Table 10, we can see that the highest value of F1 (0.698) is obtained for the ensemble configuration. This ensemble uses the FFNN component with CUI

**Table 10. Sentence fact-checking binary classification results.**

| Classifier Type | InputTr1/Tr2 | CUIs | Hidden L. | Avg. Acc. | Avg. Prec. | Avg. Rec. | Avg. F1 |
|---|---|---|---|---|---|---|---|
| (1) Ens. FFNN-Tr1(blu)-Tr2(blu) | text/fspo | true | 1 | 0.701±0.009 | 0.701±0.007 | 0.709±0.007 | **0.698±0.008** |
| (2) Ens. FFNN-Tr1(blu)-Tr2(blu) | text/fspo | false | 1 | 0.697±0.010 | 0.699±0.008 | 0.706±0.008 | 0.695±0.009 |
| (3) Ens. Tr1(blu)-Tr2(blu) | text/fspo | - | 1 | 0.697±0.011 | 0.698±0.008 | 0.705±0.008 | 0.694±0.010 |
| (4) Transformer (blu) | text | false | - | 0.693±0.016 | 0.688±0.012 | 0.691±0.015 | 0.686±0.015 |
| (5) Ens. Tr1(alb)-Tr2(blu) | text/fspo | - | 1 | 0.684±0.014 | 0.683±0.019 | 0.689±0.020 | 0.680±0.017 |
| (6) Ens. Tr1(alb)-Tr2(blu) | fspo/fspo | - | 1 | 0.680±0.007 | 0.685±0.010 | 0.691±0.010 | 0.678±0.007 |
| (7) Ens. FFNN-Tr1(bio)-Tr2(blu) | vspo/fspo | true | 1 | 0.660±0.009 | 0.658±0.008 | 0.663±0.008 | 0.655±0.008 |
| (8) Transformer (blu) | fspo | true | - | 0.637±0.032 | 0.637±0.027 | 0.640±0.029 | 0.632±0.031 |
| (9) Transformer (blu) | vspo | false | - | 0.631±0.046 | 0.628±0.040 | 0.631±0.041 | 0.625±0.043 |
| (10) FFNN (tanh-100) | | false | - | 0.598±0.010 | 0.573±0.010 | 0.565±0.007 | 0.564±0.007 |
| (11) FFNN (tanh-500) | | true | - | 0.560±0.004 | 0.553±0.008 | 0.554±0.009 | 0.551±0.007 |

Results for different models sorted by Avg. F1. pre-trained model abbreviations: *ALBERT* (alb), *BioBERT* (bio), *BlueBERT* (blu).

**Table 11. Sentence fact-checking binary classification results over the positive class.**

| Classifier Type | InputTr1/Tr2 | CUIs | Hidden L. | Avg. Acc. | Avg. Prec. | Avg. Rec. | Avg. F1 |
|---|---|---|---|---|---|---|---|
| Ens. Tr1(blu)-Tr2(blu) | text/vspo | - | 2 | 0.706±0.012 | 0.733±0.014 | 0.794±0.017 | **0.762±0.008** |
| Ens. FFNN-Tr1(blu)-Tr2(blu) | text/vspo | false | 2 | 0.706±0.011 | 0.734±0.012 | 0.792±0.023 | **0.762±0.010** |
| Ens. FFNN-Tr1(blu)-Tr2(blu) | text/vspo | true | 1 | 0.706±0.006 | 0.742±0.010 | 0.776±0.022 | 0.758±0.008 |
| Transformer (fti) | vspo | true | - | 0.605±0.009 | 0.603±0.007 | 0.983±0.014 | 0.748±0.002 |
| Transformer (fti) | fspo | true | - | 0.595±0.000 | 0.595±0.000 | 1.000±0.000 | 0.746±0.000 |
| FFNN (sigmoid-100) | | false | - | 0.595±0.000 | 0.595±0.000 | 1,000±0.000 | 0.746±0.000 |
| FFNN (sigmoid-100) | | true | - | 0.595±0.000 | 0.595±0.000 | 1,000±0.000 | 0.746±0.000 |
| Ens. Tr1(alb)-Tr2(blu) | text/vspo | - | 1 | 0.680±0.009 | 0.712±0.009 | 0.777±0.005 | 0.743±0.006 |
| Ens. FFNN-Tr1(bbc)-Tr2(blu) | vspo/vspo | true | 1 | 0.674±0.010 | 0.705±0.006 | 0.776±0.018 | 0.739±0.010 |
| Ens. FFNN-Tr1(bio)-Tr2(blu) | fspo/vspo | true | 1 | 0.671±0.005 | 0.708±0.006 | 0.763±0.009 | 0.734±0.004 |
| Transformer (blu) | text | false | - | 0.693±0.016 | 0.765±0.024 | 0.702±0.054 | 0.731±0.024 |

Results for different models sorted by Avg. F1 over the positive class (label *False*). pre-trained model abbreviations: *Funnel* (fti), *BERT Cased* (bbc), *ALBERT* (alb), *BioBERT* (bio), *BlueBERT* (blu).

codes at the input, the *BlueBERT* model pre-trained in the biomedical domain in the first Transformer component whose input is the text of the sentence, the same *BlueBERT* pre-trained model in the second Transformer component with triplets of fixed size at the input formed by the description of the CUIs extracted from the sentence, and one hidden layer. The second (0.695) and third (0.694) highest values of F1 are obtained for an ensemble configuration like the above except that the second does not use CUI codes in the FFNN and the third configuration does not use the FFNN component.

Table 12 presents a statistical significance matrix where each of the models in Table 10 is compared with the rest. It has been elaborated using the F1 values calculated for 50 different random seeds. The alternative hypothesis has been set to "greater", that is, we check if the F1 values of the model that appear in the columns are higher than those that appear in the rows. Thus, we can see that there are no statistically significant differences between the first three models in Table 10. Continuing with other results in this table, the fourth highest value (F1 = 0.686) is obtained by a Transformer model pre-trained in the biomedical domain (*BlueBERT*) that receives as input the text of the sentence. In this case, the difference is significant

**Table 12. Sentence fact-checking statistical significance matrix.**

| Cl. Type | (1) | (2) | (3) | (4) | (5) | (6) | (7) | (8) | (9) | (10) |
|---|---|---|---|---|---|---|---|---|---|---|
| (2) | **0.11237** | | | | | | | | | |
| (3) | **0.05304** | **0.33596** | | | | | | | | |
| (4) | 0.00000 | 0.00000 | 0.00000 | | | | | | | |
| (5) | 0.00000 | 0.00000 | 0.00000 | **0.32361** | | | | | | |
| (6) | 0.00000 | 0.00000 | 0.00000 | **0.08001** | 0.21932 | | | | | |
| (7) | 0.00000 | 0.00000 | 0.00000 | 0.00000 | 0.00000 | 0.00000 | | | | |
| (8) | 0.00000 | 0.00000 | 0.00000 | 0.00000 | 0.00000 | 0.00000 | 0.00001 | | | |
| (9) | 0.00000 | 0.00000 | 0.00000 | 0.00000 | 0.00000 | 0.00000 | 0.00005 | **0.17609** | | |
| (10) | 0.00000 | 0.00000 | 0.00000 | 0.00000 | 0.00000 | 0.00000 | 0.00000 | 0.00000 | 0.00036 | |
| (11) | 0.00000 | 0.00000 | 0.00000 | 0.00000 | 0.00000 | 0.00000 | 0.00000 | 0.00000 | 0.00026 | 0.00000 |

The numbers in row and column headers correspond to the models in Table 10, and for each pair of models the p-value obtained by means of a Wilcoxon test is shown.

compared to the first three models. Then appears the first ensemble model (F1 = 0.680) whose first Transformer component uses a pre-trained model in the general domain (*ALBERT*), fed with the text of the sentence. This ensemble does not incorporate a FFNN component. The following model (F1 = 0.678) is similar but the Transformer component is fed with SPO triplets of fixed lenghts formed with the descriptions of the CUIs. These last three mentioned models are at the same level of performance with respect to the F1 values. Below is the first ensemble that uses SPO triplets of variable length in the first Transformer component (F1 = 0.655). In this case the Transformer has been pre-trained in the biomedical domain (*BioBERT*). Next (F1 = 0.632) appears the first Transformer that uses fixed-length SPO triplets created with the descriptions of the CUIs, and the Transformer (F1 = 0.625) that uses variable-length SPO triplets created from the text of the sentence. Both models have been pre-trained in the biomedical domain (*BlueBERT*). The difference between the F1 values of these two models is not significant. And finally with a considerably greater difference with respect to Transformers and ensembles appear the FFNN that has as input TF-IDF vectors extracted from the text of the sentence (F1 = 0.564), and the FFNN whose input is formed by vectors TF-IDF created from CUI codes (F1 = 0.551).

In general, if we look at the statistical significance matrix, we can detect two groups of models with similar performances. The first would be made up of the ensembles that use the *BlueBERT* model. The second group would be formed by the *BlueBERT* Transformer model along with the ensembles that use pre-trained Transformers in the general domain. Behind these groups appear the rest of the evaluated configurations.

If we now review the values of F1 with respect to the positive class (Table 11), we can see that the differences between the F1 values are smaller (0.762 to 0.731). Although the highest values are still obtained with the ensemble models whose Transformer components have been pre-trained in the biomedical domain. We also see that the values of F1 are higher than in Table 10 (calculated using macro average), due to the existing class distribution of the factual and relevant sentences that make the positive class (label *False*) the majority (Table 2).

## 6.3 Fake news classification results

The fake-news classification has been carried out with the news items that contain some factual relevant sentence, with the same *training-test* partition as the one used in the fact-checking classification: 170 news item instances in the *training* dataset, 43 news item instances in the *dev/validation* dataset, and 61 news item instances in the *test* dataset. The distribution of instances by class (*training+dev/validation*: F = 63.38%, T = 36.62%; *test*: F = 73.77%, T = 26.23%) is dictated by the partition of sentences previously made in the fact-checking classification process.

We have proceeded to classify these news items using the rule described in the section 5.2.4. To contextualize the results with a fairly demanding baseline, we have also classified these articles using different Transformer models, feeding their inputs with the first *n* tokens of the article's full text, where *n* is the configured sequence size. Table 13 shows the averaged results for this fake news classification.

As we can see in Table 13, the two highest values of F1 (0.747, 0.726) are obtained with Transformer models that have been pre-trained in the general domain (*BERT Cased*, *ALBERT*). The third highest value (F1 = 0.716) is for *BioBERT*, a Transformer model that has been pre-trained in the biomedical domain. Our proposal based on the fact-checking classification of sentences scores the fourth-highest F1 value (0.703), ahead of most Transformer models.

**Table 13. Fake news detection binary classification results.**

| Classifier Type | Avg. Acc. | Avg. Prec.+ | Avg. Rec.+ | Avg. F1+ | Avg. Prec. | Avg. Rec. | Avg. F1 |
|---|---|---|---|---|---|---|---|
| BERT Cased (200) | 0.792±0.035 | 0.889±0.026 | 0.822±0.056 | 0.853±0.028 | 0.742±0.041 | 0.764±0.037 | 0.747±0.037 |
| ALBERT (250) | 0.784±0.030 | 0.864±0.021 | 0.840±0.064 | 0.851±0.026 | 0.734±0.047 | 0.733±0.029 | 0.726±0.027 |
| BioBERT (200) | 0.790±0.038 | 0.869±0.048 | 0.851±0.075 | 0.856±0.027 | 0.708±0.124 | 0.735±0.085 | 0.716±0.107 |
| **Sentence-based (best seed)** | 0.770 | 0.844 | 0.844 | 0.844 | 0.703 | 0.703 | 0.703 |
| BERT Uncased (150) | 0.751±0.048 | 0.867±0.016 | 0.785±0.088 | 0.821±0.044 | 0.705±0.051 | 0.720±0.022 | 0.702±0.036 |
| Funnel (200) | 0.767±0.015 | 0.864±0.051 | 0.822±0.070 | 0.839±0.010 | 0.680±0.110 | 0.718±0.081 | 0.693±0.095 |
| **Sentence-based** | 0.746±0.028 | 0.840±0.044 | 0.809±0.049 | 0.824±0.016 | 0.679±0.115 | 0.689±0.080 | 0.682±0.098 |
| BlueBERT (200) | 0.728±0.046 | 0.845±0.017 | 0.774±0.078 | 0.806±0.042 | 0.672±0.038 | 0.687±0.028 | 0.672±0.036 |
| ClinicalBERT (200) | 0.764±0.057 | 0.853±0.080 | 0.847±0.137 | 0.838±0.048 | 0.630±0.188 | 0.689±0.133 | 0.645±0.159 |

Results for different models sorted by Avg. F1. In boldface are shown the results of our classifier based on the veracity of the sentences. The other models use the full news item as input (the maximum sequence size is shown in parentheses). In addition to accuracy, the table shows the values of precision, recall, and F1 measure for the positive class (label *False*), and using macro average.

These results are not surprising since Transformers usually reach results that are in the state of the art [76] when the task of detecting fake news uses as input full articles without any additional information. However, the results obtained with our simple rule-based classifier are not so far from those results, and even surpass some of the Transformer models. In addition, with our proposal we have the advantage of offering an additional level of explainability by providing truth evaluation at the sentence level.

F1 values obtained for the positive class (label *False*) are higher than when we consider the macro average of the two classes as it happens in the fact-checking classification of sentences. This is because the articles have been collected from sources in which the *False* and *Partially False* classes predominate. On the other hand, we can also verify that the highest values of F1 + are obtained with Transformer models whose input is made up of the first *n* tokens of the article.

To check the behavior of the system with news items external to our corpus, we have created an additional test dataset with the articles contained in the *ReCOVery* repository [81], also related to the field of health.

By maintaining the check-worthy and fact-checking sentence classification models generated with our training dataset, we have obtained 1276 factual relevant sentences associated with 596 articles. On those 596 articles we have executed both our rule-based classifier (5.2.4), and the Transformer models trained with full articles of our corpus.

The results can be seen in Table 14, where we have also included those published by the authors of this corpus. These results are not directly comparable because we do not know which instances were in their test dataset and we have only used articles that had some factual relevant sentences, but can give an idea of the behavior of our system. It should be noted that our system is trained with 170 news items of our corpus and has been evaluated with 596 *ReCOVery* articles, while according to the authors, their tests have been done with 1623 training instances and 406 test instances. Despite this initial disadvantage, our proposal is capable of obtaining an F1 value of 0.652 for the best random seed. Ahead of most of the Transformer models and two of the models evaluated by the authors of this repository.

## 6.4 Full pipeline classification results

The previous results are based on the premise of obtaining the best automatic annotations for the new articles that are incorporated into the corpus. Therefore the fact-checking sentence

**Table 14. Fake news detection binary classification (*ReCOVery* dataset).**

| Classifier Type | Avg. Acc. | Avg. Prec.+ | Avg. Rec.+ | Avg. F1+ | Avg. Prec. | Avg. Rec. | Avg. F1 |
|---|---|---|---|---|---|---|---|
| *ReCOVery—SAFE* | | | | | 0.752 | 0.753 | 0.753 |
| BlueBERT (250) | 0.679±0.035 | 0.542±0.048 | 0.667±0.059 | 0.594±0.023 | 0.667±0.025 | 0.676±0.021 | 0.663±0.028 |
| *ReCOVery—LIWC+DT* | | | | | 0.660 | 0.662 | 0.660 |
| Funnel (200) | 0.676±0.136 | 0.568±0.137 | 0.787±0.112 | 0.642±0.072 | 0.664±0.182 | 0.701±0.088 | 0.657±0.155 |
| **Sentence-based (best seed)** | **0.663** | **0.516** | **0.690** | **0.591** | **0.655** | **0.669** | **0.652** |
| ALBERT (150) | 0.665±0.061 | 0.537±0.083 | 0.637±0.086 | 0.574±0.033 | 0.656±0.045 | 0.658±0.038 | 0.646±0.049 |
| **Sentence-based** | **0.645±0.009** | **0.498±0.008** | **0.735±0.026** | **0.593±0.005** | **0.652±0.004** | **0.665±0.004** | **0.639±0.007** |
| *ReCOVery—Text-CNN* | | | | | 0.634 | 0.627 | 0.630 |
| BERT Uncased (250) | 0.591±0.045 | 0.450±0.028 | 0.657±0.115 | 0.529±0.036 | 0.601±0.021 | 0.606±0.027 | 0.579±0.042 |
| *ReCOVery—RST+DT* | | | | | 0.571 | 0.573 | 0.571 |
| BioBERT (250) | 0.572±0.092 | 0.442±0.057 | 0.619±0.152 | 0.504±0.028 | 0.547±0.134 | 0.583±0.044 | 0.547±0.108 |
| BERT Cased (250) | 0.531±0.067 | 0.412±0.037 | 0.720±0.071 | 0.521±0.025 | 0.571±0.041 | 0.574±0.043 | 0.526±0.067 |
| ClinicalBERT (200) | 0.425±0.071 | 0.340±0.026 | 0.682±0.242 | 0.444±0.065 | 0.385±0.147 | 0.484±0.032 | 0.390±0.099 |

Results for different models sorted by Avg. F1. In boldface are shown the results of our binary classifier based on the veracity of the sentences. In italics, the results published with the *ReCOVery* dataset that are based on a different training/evaluation process. The other models use the full news item as input. In addition to accuracy, the table shows the values of precision, recall, and F1 measure for the positive class (label *False*), and using macro average.

classification has been trained with the factual relevant sentences in which this check-worthiness class value has been manually annotated.

In this section, on the contrary, we wanted to analyze the results obtained when the fact-checking training is carried out with the results automatically annotated by the check-worthiness classifier. That is, there will be a propagation of the errors made in the first stage to the second stage of classification.

To do this, we have selected the Transformer model with the best absolute results (best random seed) in F1 measure on the positive class, that is, the one that indicates that the sentence is factual and relevant. Thus, we have selected the Funnel model with which we have obtained an F1 measure on the positive class of 0.619 and an average F1 measure on the two classes of 0.779. With this model we have performed the check-worthiness annotation in the test set. On the sentences marked as factual relevant, the following stages of sentence processing have been carried out without any supervision (extraction of SPOs, CUIs, etc.) until a total of 1145 factual and relevant sentences corresponding to 262 news items are obtained. Based on these sentences and following the criteria described in Section 5.2.3, we performed a grid search to determine which classification model was the most suitable to perform the automatic fact-checking classification. Finally, we have selected an ensemble model composed of an FFNN component with *TF-IDF* vectors formed by the sentence text, a *BlueBERT Transformer* component whose input is the sentence text, and a second *BlueBERT Transformer* component whose input is composed of fixed-length SPO triples generated from the descriptions of the CUIs. This ensemble model has obtained an average F1 measure of 0.684 and an average F1 measure over the positive class of 0.713. We have carried out the fact-checking annotation of sentences with this model. Finally, similar to what is explained in section 5.2.4, we have evaluated the 262 selected news items to determine their veracity using Transfomer models, and our sentence-based classifier. This has been done both for the test set of our dataset (53 news items) and for the external *ReCOVery* corpus (595 news items). The results obtained are shown respectively in Tables 15 and 16.

**Table 15. Full pipeline fake news detection binary classification results.**

| Classifier Type | Avg. Acc. | Avg. Prec.+ | Avg. Rec.+ | Avg. F1+ | Avg. Prec. | Avg. Rec. | Avg. F1 |
|---|---|---|---|---|---|---|---|
| Funnel (250) | 0.821±0.032 | 0.967±0.024 | 0.785±0.054 | 0.865±0.030 | 0.789±0.021 | 0.853±0.026 | 0.798±0.030 |
| BioBERT (150) | 0.819±0.027 | 0.921±0.035 | 0.828±0.062 | 0.870±0.024 | 0.778±0.026 | 0.811±0.041 | 0.783±0.030 |
| ALBERT (250) | 0.819±0.020 | 0.889±0.034 | 0.864±0.042 | 0.875±0.015 | 0.771±0.023 | 0.778±0.046 | 0.770±0.034 |
| BERT Uncased (250) | 0.817±0.027 | 0.885±0.054 | 0.869±0.044 | 0.875±0.015 | 0.768±0.035 | 0.771±0.073 | 0.762±0.054 |
| BlueBERT (150) | 0.768±0.047 | 0.879±0.048 | 0.803±0.106 | 0.833±0.046 | 0.726±0.038 | 0.737±0.062 | 0.713±0.059 |
| BERT Cased (200) | 0.778±0.039 | 0.863±0.058 | 0.839±0.077 | 0.847±0.029 | 0.721±0.052 | 0.723±0.090 | 0.706±0.083 |
| **Sentence-based** | 0.734±0.006 | 0.819±0.007 | 0.821±0.000 | 0.820±0.003 | 0.657±0.009 | 0.657±0.011 | 0.657±0.010 |
| Clinical BERT (250) | 0.777±0.039 | 0.838±0.085 | 0.890±0.097 | 0.855±0.021 | 0.636±0.187 | 0.677±0.141 | 0.641±0.161 |

Results for different models ordered by Avg. F1 after executing the complete pipeline, in which the results predicted by the check-worthy classifier are used in the next fact checking classification stage. In boldface are shown the results of our classifier based on the veracity of the sentences. The other models use the full news item as input (the maximum sequence size is shown in parentheses). In addition to accuracy, the table shows the values of precision, recall, and F1 measure for the positive class (label *False*), and using macro average.

As we can see in the first of these tables, in the test set our sentence-based model achieves results very close (F1 = 0.657) to those obtained when we use the manual check-worthy annotation (F1 = 0.682). These results are reasonably good considering the difficulty of the problem, and are expected to improve as the size of the dataset increases. On the other hand, on the *ReCOVery* dataset, which has a much larger number of instances to evaluate, our model is competitive with an average value of F1 (0.661) higher than that obtained by most Transformer models.

## 6.5 Error analysis

To better understand the results obtained, we have extracted the annotations made by the classifiers in each of the stages of the pipeline, analysing the errors made in search of any pattern that can give clues about the causes of these errors. We have based this on the results of the full pipeline presented in section 6.4. From now on, when we talk about error rate, we are referring to the percentage of sentences incorrectly annotated by the classifier with respect to our annotation carried out manually.

**Table 16. Full pipeline fake news detection binary classification (*ReCOVery* dataset).**

| Classifier Type | Avg. Acc. | Avg. Prec.+ | Avg. Rec.+ | Avg. F1+ | Avg. Prec. | Avg. Rec. | Avg. F1 |
|---|---|---|---|---|---|---|---|
| Funnel (250) | 0.786±0.068 | 0.704±0.114 | 0.744±0.093 | 0.713±0.062 | 0.779±0.059 | 0.776±0.054 | 0.769±0.063 |
| **Sentence-based** | 0.676±0.004 | 0.532±0.005 | 0.664±0.008 | 0.590±0.004 | 0.660±0.003 | 0.673±0.003 | 0.661±0.004 |
| ALBERT (250) | 0.633±0.073 | 0.500±0.070 | 0.762±0.085 | 0.597±0.033 | 0.658±0.036 | 0.662±0.043 | 0.625±0.066 |
| BERT Uncased (250) | 0.602±0.063 | 0.468±0.048 | 0.778±0.097 | 0.579±0.041 | 0.639±0.035 | 0.642±0.044 | 0.596±0.061 |
| BlueBERT (200) | 0.544±0.102 | 0.415±0.048 | 0.601±0.198 | 0.476±0.047 | 0.547±0.056 | 0.557±0.047 | 0.511±0.123 |
| BioBERT (250) | 0.517±0.063 | 0.398±0.036 | 0.673±0.077 | 0.496±0.020 | 0.550±0.033 | 0.553±0.037 | 0.511±0.058 |
| BERT Cased (250) | 0.504±0.103 | 0.399±0.045 | 0.689±0.207 | 0.489±0.041 | 0.563±0.057 | 0.547±0.043 | 0.475±0.118 |
| Clinical BERT (250) | 0.391±0.063 | 0.341±0.018 | 0.775±0.178 | 0.468±0.039 | 0.391±0.121 | 0.479±0.028 | 0.354±0.089 |

Results for different models ordered by Avg. F1 after executing the complete pipeline, in which the results predicted by the check-worthy classifier are used in the next fact checking classification stage. In boldface are shown the results of our binary classifier based on the veracity of the sentences. In italics, the results published with the *ReCOVery* dataset that are based on a different training/evaluation process. The other models use the full news item as input. In addition to accuracy, the table shows the values of precision, recall, and F1 measure for the positive class (label *False*), and using macro average.

In the check-worthy classification of sentences carried out in the first stage, 10.42% of the sentences were scored incorrectly in the test set. If we take into account the veracity of the article to which they belong, there are no major differences in this percentage (10.55% when the article is false and 10.36% when the article is true). The average length of correctly classified sentences is 144 characters while the average length of incorrectly classified sentences is 114, which may indicate that the additional information that longer sentences have is beneficial to the classification process. Taking into account only the sentences that are factual and relevant, those that are false have an error rate of 26.76% while in those that are true this error rate is 35.79%. This indicates that true sentences lead to greater error in check-worthy classification, possibly because false sentences tend to contain more controversial statements that attract the reader's attention, so they can be more easily detectable than true sentences.

Regarding fact-checking classification, 28.69% of factual and relevant sentences have been annotated incorrectly. Whether the article they belong to is true or false also does not have a great influence on the error rate (28.57% and 28.92% respectively). The average length of correctly annotated sentences is 143 characters, while that of incorrectly annotated sentences is 145 characters. The presence of medical terms in the sentence slightly increases the error rate (30.95%) compared to that of sentences containing more common terms (28.21%). The only aspect that seems to indicate a significant influence on the error rate is whether the predicate is one of those we have selected (Section 9) as an action in the field of health (28.83%), or is another type of predicate that is more common (43.40%). The first type of sentences may be benefiting from the health-trained transformers models we use in the selected ensemble to annotate these sentences.

## 7 Discussion

The results obtained by the Transformer and ensemble models in the fact-checking classification of sentences (Table 10) are clearly superior to traditional NLP approaches such as FFNNs that use TF-IDF vectors as input. In addition, within these Transformer models, *BlueBERT* [65] obtains a significant advantage over the rest. This model, based on a *BERT Base Uncased*, has been pre-trained with abstracts from the *PubMed abstract* corpus (4,000M words) and with clinical notes from the *MIMIC-III* corpus (500M words). It therefore makes sense that when combining general information and information from the biomedical domain, it behaves better than the rest, since the documents contained in our corpus are mainly extracted from blogs, social networks and general newspapers, although with some medical terms. It also indicates that it is possible to extract this implicit knowledge captured during pretraining and use it to verify the veracity of sentences without the need to resort to traditional knowledge bases (RQ1).

On the other hand, we also observe that when we combine within an ensemble model, a Transformer that uses the text of the sentence, with a second Transformer that uses SPO triplets (syntactic information) with the description of the CUIs extracted from the sentence (biomedical information) we obtain higher F1 values than when we used that first Transformer model alone (RQ2, RO1). This indicates that the use of this type of biomedical concepts extracted from the plain text of the sentence delivered in a structured way at the input of the model provides additional information that is useful when determining the veracity of a sentence. This is also favored by the fact of using pre-trained models in the biomedical domain, since it increases the chances that these CUI descriptions have been included in any of the documents used in the pre- training process. In Table 12 we can verify that these performance differences between our proposal and the best Transformer model are statistically significant.

Starting from the ensemble model for the fact-checking classification of sentences with the highest F1 value (Table 10), the fake-news classification of complete articles has been carried out using the rule described in Section 5.2.4. As we can see in Table 10, the three highest F1 values are obtained by Transformer models that use between 200 and 250 news item tokens to determine their veracity. This, as previously mentioned, is not surprising since these models tend to be among those that obtain the best results in this task of classifying full articles. Two of them have been pre-trained in the general domain, which indicates that using pre-trained models in the biomedical domain is not decisive, at least for this task. Our proposal is in fourth position with a value of F1 (0.703), not too far from that obtained by any of these Transformer models. It is a value high enough to evaluate new articles, which allows us to use it as the last stage of the process, always considering that the result obtained is an estimate and should be subsequently verified by a human evaluator.

When we have evaluated these models with the external *ReCOVery* repository, the results have been similar. In this case, our proposal has obtained the fifth highest value of F1, behind two Transformer models and two of the models presented by the authors of this repository. The F1 values have been somewhat lower than those obtained with our corpus. *ReCOVery* is made up of articles related to COVID19 collected from websites similar to those used in our work. So it is likely that this lower performance is due to the fact that the number of instances of our corpus used for training (170) is quite less than the number of *ReCOVery* instances used to perform the evaluation (596). The results obtained by the Transformer models are reasonably high despite this disadvantage. However, one of its main drawbacks is that these results are difficult to interpret. On the other hand, our sentence-based proposal, once the veracity of an article has been evaluated, allows access to the individual evaluation of its sentences. Consulting these sentences labeled with their check-worthiness and their truth value, we obtain a justification of the veracity of the complete article (RQ3). This also helps a user to have more confidence in the results provided by the system when evaluating a piece of news.

The results of this work have shown that it is possible to permanently monitor the news that appears on the selected websites and decide if it is fake news with minimal supervision and high accuracy (RO2). The only supervision required is the initial acceptance of the article and completion of the text if it could not be captured correctly, and the confirmation (or correction if necessary) of the predicate proposed by the system for each sentence. The remaining components of the pipeline can operate completely automatically, providing a classification result of each sentence and each news based on them. In addition, the system trained with an initial set of medical news can be periodically retrained as new classified news accumulates. In this way the system can improve its behavior and adapt to new forms of disinformation.

When unobserved instances are evaluated, although the system can generalize from the knowledge implicit in Transformer models and the one acquired after fine-tuning made on our own corpus, it is possible that over time its performance will be reduced. However, using newer versions of Transformer models and retraining it back over the corpus including the latest news incorporated therein, it should restore its effectiveness.

Since one of the objectives of the work is focused on the improvement of fact-checking models of sentences, in the fact-checking and fake news detection, the evaluation have been carried out using gold labels, that we have obtained applying the proposed system in attended mode. However, in the unattended functioning of the system, when the unobserved news items are evaluated, a check-worthiness classification is carried out and the sentences labeled as factual and relevant are selected for the fact-checking classification. This means that the errors of the first classification process are propagated to the second, although the final results at the article level are not particularly affected, as can be seen in Table 14. To confirm this, we have done an additional experiment (Section 6.4), in which the results of the check-worthiness

classifier are used to select the sentences to pass to the next stages of the pipeline. The results obtained in the fake news classification by our sentence-based model indicate that, despite the possible propagation of errors from the initial stage, its performance remains competitive (Table 16).

Another limitation is that only news items that have at least one factual and relevant sentence can be evaluated and in general, most of the sentences in a news item are non-relevant or non-factual. Nevertheless, most of the articles used (84%) contain at least one factual and relevant sentence, even considering that some sentences can be ignored if they use subjective or not very assertive language.

## 8 Conclusions

In this article we present an end-to-end framework that allows the annotation and classification of sentences according to their check-worthiness and their truth value. Based on this sentence classification process, the framework carries out a fake news classification of the news items that group these sentences.

We have developed a method for extracting subject-predicate-object triplets from the sentence text. We have also developed an ensemble classifier composed of an FFNN and two Transformer models that supports as input the triples mentioned above, text sequences, and CUI medical identifiers. This architecture has allowed us to experiment with different combinations of these types of inputs confirming that when the sentence text sequence is used in combination with triplets generated from the description of the CUIs, our ensemble model has a higher performance than any Transformer model working alone.

In the fake news classification, we wanted to evaluate the feasibility of using sentence information to determine the truth value of complete news items, detecting that although the results obtained with our heuristics based on simple rules (F1: 0.703) are somewhat inferior to those obtained by the best Transformer models (F1: 0.747), when we expose it at the news items of a bigger corpus such as *ReCOVery* the performance remains competitive (F1: 0.652) and higher than most Transformer models analyzed. In addition, the use of this strategy provides us with an additional level of justification if we consult the fact-checking prediction of sentences, which is not present in Transformer models.

Finally, in this work we have compiled a corpus oriented to the detection of fake news in the health domain composed of 327 articles and a total of 10335 sentences that are annotated both in their check-worthiness and their truth value. This level of detail provided by sentence-level annotation distinguishes it from other published corpora where annotation is done only at the article level.

### 8.1 Future work

Looking forward, we plan to continue expanding the dataset with new sources, thus increasing the number of news items captured periodically and boosting redundancy. We also want to explore the possibility of integrating multimodal information into the corpus. Additionally, we aim to enhance the triplet extraction process to get more than one triplet per sentence. We hope that these triplets will be simpler than the current ones, which would enable us to identify and store recurring triplets to be used in new classification models. Another aspect that would be interesting to study is the correction or discarding of erroneous CUI mappings that can lead to biases in the classification. Finally, we also plan to look at the effect of unattended processing of an external biomedical dataset and use the resulting annotations to retrain our models.

## Appendix 1

**Check-worthiness annotation criteria**. To classify a sentence as non-factual we have considered the following cases:

- Promises about what someone is going to do in the future and therefore cannot be checked at this time: *"I will no longer vaccinate my other children."*.

- Proverbs and sayings: *"Prevention is better than cure."*. These expressions are not usually used to indicate a fact.

- Instructions given/received and recipes, do not imply a statement about some fact but a sequence of steps to be performed: *"2.Use a 24k-14k Gold ring to scratch on the lipstick."*.

- Affirmations about one's own emotions and mood: *"I opened one of his eyes and I just knew inside my heart that it was something really bad."*. They are subjective phrases that cannot be checked.

- Opinions and value judgments: *"I think I would call death an adverse side-effect"*. Since personal opinions cannot be disapproved of they are considered non-factual.

- Questions that do not contain a claim in themselves: *"What is Crohn's Disease?"*.

- Vague predictions that cannot be checked: *"We are committed to transforming over a billion lives. . .and we won't stop there!"*.

And to classify a sentence as factual we have followed these criteria, which also include evaluating the relevance of the sentence to be factually verified:

- Own emotions and mood that contain affirmations about some fact: *"As a physician, I feel very bad because I believe our vaccine is a good treatment that can extend these people's lives."* In this case, the sentence talks about emotions and contains a fact, but it is not very specific. In addition, they will be relevant if they are interesting to a wide audience: *"I feel bad because I had heard that the MMR vaccine causes autism and yet I let my son get it."*.

- Opinions and value judgments that contain statements about some fact: *"In practice, not all pregnant women receive flu shots, and I think that universal vaccination of pregnant women could get us into a whole new set of problems."*. Statement *"not all pregnant women receive flu shots"* is not particularly controversial. In addition, they will be relevant if they are interesting to a wide audience: *"I consider it unacceptable that the number of people vaccinated for COVID does not exceed 10% in the countries of the African continent while in Europe the vaccines expire."*.

- Questions that contain statements about some fact: *"Olive oil comes from olives, peanut oil from peanuts, sunflower oil from sunflowers; but what is a canola?"*. In addition, they will be relevant if they are interesting to a wide audience: *"Did you know that the government earns royalties from the sale of the Gardasil vaccine?"*.

- Specific predictions that can be checked. They will be relevant if they are interesting to a wide audience: *"The more people get vaccinated, the less severe this Omicron outbreak will be."*.

- Claims in sentences containing references to reports and investigations can be checked by accessing these documents. In addition, they will be relevant if they are interesting to a wide audience: *"It has been reported by the US National Cancer Institute, that asparagus is the*

*highest tested food containing glutathione, which is considered one of the body's most potent anticarcinogens and antioxidants.".*

- Procedures and announcements indicating how certain facts are to develop, are considered factual but not relevant: *"At your first vaccination appointment, you should get a CDC COVID-19 Vaccination card that tells you what COVID-19 vaccine you received, the date you received it, and where you received it.".*

- Vague statements in which it is not clear who the subject is, or it is not very clear what they mean: *"Prolonged exposure will cause Leukemia, increasing the risk of cancer.".* Regarding this criterion we have considered the sentence as a unit of annotation, therefore it must contain everything necessary to be checked without resorting to other sentences of the news item.

- Performative sentences, i.e. sentences that become true just by being pronounced, are always considered factual and not relevant: *"I also want to say a word to parents: If your children are not vaccinated, please get them vaccinated.".*

- Phrases in which a claim includes figures and numbers are always considered factual. In addition, they will be irrelevant if the claim about these figures is obvious, non-controversial, or corresponds to official and public information: *"In the United States, about 44 per cent of the population is type O, while about 41 per cent is type A.".* On the other hand, it will be considered relevant if those figures indicate a change, are controversial or with values out of the ordinary: *"Never mind that the Gardasil vaccine is responsible for ending the lives of 271 young women to date, according to over 57,520 adverse event reports obtained from the Vaccine Adverse Events Reporting System.".*

- Sentences with quotes about what another person has said are always factual: *""Measles is incredibly infectious," Mazer said.".* In fact in this type of sentence there are two claims, the statement about the quote and the quote itself. In general, we have followed the criterion of using the quotation as the main claim to consider when determining the type of sentence. As in previous cases if it is interesting for a wide audience it will be relevant: *"The Delta variant makes it easier for vaccinated people to transmit the virus, the CDC said.".*

- And in general, sentences containing statements about some fact: *"Austrian biologist Karl Landsteiner discovered the main blood groups in 1901, naming them type A, B, AB and O.".* In addition, they will be relevant if they are interesting to a wide audience: *"Since cancer is only a deficiency of vitamin B17, eating 15 to 20 pieces of apricot stone/nucleus (fruit stone) everyday is enough.".*

## Appendix 2

**UMLS semantic groups**. Semantic groups used as a filter in the extraction of CUIs.

- Activities & Behaviors

- Anatomy

- Chemicals & Drugs

- Concepts & Ideas

- Devices

- Disorders

- Genes & Molecular Sequences

- Geographic Areas

- Living Beings

- Objects

- Occupations

- Organizations

- Phenomena

- Physiology

- Procedures

## Appendix 3

**Verbs used for SPO triplet extraction**. The verbs listed below and their different conjugations have been used to extract subject, predicate and object from the factual and relevant sentences. The objective of using them is to give preference to capturing triplets related to diseases, causes, symptoms, treatments, and changes in vital signs.

- **Verbs related to diseases, causes or symptoms**: Causing, spreading, acquiring, activating, affecting, producing, attacking, developing, inducing, damaging, leading, promoting, transmitting, triggering, yielding, hurting, harming, disrupting, resulting, contributing, dying, experiencing, infecting, releasing, associating, showing, providing, recovering, contaminating, predicting, diagnosing, linking, injuring, suggesting, banning, living, making, feeding, working, acting, blowing, reflecting, noticing.

- **Verbs related to treatments or medications**: Improving, helping, preventing, killing, eliminating, fighting, protecting, avoiding, combating, blocking, curing, benefiting, destroying, stopping, neutralizing, healing, truncating, relying, reversing, treating, remedying, inhibiting, hampering.

- **Verbs that indicate changes in vital signs or other measured variables**: Increasing, reducing, slowing, decreasing, lowering, boosting, amplifying, fueling, fuelling, elevating, exacerbating, raising, rising, quieting, tripling, doubling, varying, halving.

- **Other frequent verbs in the medical domain**: Containing, contradicting, keeping, gaining, receiving, becoming, prescribing.

## Appendix 4

**Hardware used**. The development and validation of the models described in this article has been carried out on a Celsius R940 workstation with an Nvidia Tesla M40 24GB graphics accelerator and a Celsius W520 workstation with an Nvidia GeForce RTX 3060 graphics card. Agents running on background have been deployed on a dedicated Dell OptiPlex 5050 Micro computer with 16 GB of RAM where each element (Postgres SQL Server, dataset agent, web server/Django) has been virtualized using Linux Containers.

## Supporting information

**S1 Data. The dataset, with sentence-level annotation information, is available at the link https://doi.org/10.5281/zenodo.10802196.**
(DOCX)

**S2 Data.** The source code is available in the following repositories:

- Backend: https://github.com/jrmtnez/hnfc-agent

- Frontend: https://github.com/jrmtnez/hnfc-site
  (DOCX)

## Author Contributions

**Conceptualization:** Juan R. Martinez-Rico, Lourdes Araujo, Juan Martinez-Romo.

**Data curation:** Juan R. Martinez-Rico, Lourdes Araujo, Juan Martinez-Romo.

**Software:** Juan R. Martinez-Rico.

**Supervision:** Lourdes Araujo, Juan Martinez-Romo.

**Validation:** Juan R. Martinez-Rico, Lourdes Araujo, Juan Martinez-Romo.

**Writing – original draft:** Juan R. Martinez-Rico.

**Writing – review & editing:** Lourdes Araujo, Juan Martinez-Romo.

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
