## [Decision Letter · Decision Letter 0]

1 Feb 2024

PONE-D-23-40818Building a framework for fake news detection in the health domainPLOS ONE

Dear Dr. Martinez-Rico,

Thank you for submitting your manuscript to PLOS ONE. After careful consideration, we feel that it has merit but does not fully meet PLOS ONE’s publication criteria as it currently stands. Therefore, we invite you to submit a revised version of the manuscript that addresses the points raised during the review process.

As you can see, both the reviewers have raised many concerns, especially of a methodological nature, to which I invite you to pay attention. The article in its current version needs substantial revision

We look forward to receiving your revised manuscript.

Kind regards,

Ramona Bongelli, Ph.D.

Academic Editor

PLOS ONE

Journal Requirements:

2. In your Methods section, please include additional information about your dataset and ensure that you have included a statement specifying whether the collection and analysis method complied with the terms and conditions for the source of the data.

"INITIALS: LAS, JMR

GRANT NUMBER: PID2019-106942RB-C32

FUNDER: Spanish Ministry of Science and Innovation

URL FUNDER: https://www.ciencia.gob.es/

INITIALS: JMR, LAS

GRANT NUMBER: TED2021-130398B-C21

FUNDER: Spanish Ministry of Science and Innovation

URL FUNDER: https://www.ciencia.gob.es/

INITIALS: JMR, LAS

GRANT NUMBER: PID2022-136522OB-C21

FUNDER: Spanish Ministry of Science and Innovation

URL FUNDER: https://www.ciencia.gob.es/

INITIALS: LAS

GRANT NUMBER: RAICES (IMIENS 2022)

FUNDER: IMIENS (Instituto Mixto de Investigación-Escuela Nacional de Sanidad)

URL FUNDER: " ext-link-type="uri" xlink:type="simple">https://www.imiens.es/"

"This work has been partially supported by the Spanish Ministry of Science and Innovation within the DOTT-HEALTH Project (MCI/AEI/FEDER, UE) under Grant PID2019-106942RB-C32, OBSER-MENH Project (MCIN/AEI/10.13039/501100011033

and NextGenerationEU”/PRTR) under Grant TED2021-130398B-C21, and EDHER-MED under grant PID2022-136522OB-C21 as well as project RAICES (IMIENS 2022)."

"This work has been partially supported by the Spanish Ministry of Science and Innovation within the DOTT-HEALTH Project (MCI/AEI/FEDER, UE) under Grant PID2019-106942RB-C32, OBSER-MENH Project (MCIN/AEI/10.13039/501100011033

and NextGenerationEU”/PRTR) under Grant TED2021-130398B-C21, and EDHER-MED under grant PID2022-136522OB-C21 as well as project RAICES (IMIENS 2022)."

"INITIALS: LAS, JMR

GRANT NUMBER: PID2019-106942RB-C32

FUNDER: Spanish Ministry of Science and Innovation

URL FUNDER: https://www.ciencia.gob.es/

INITIALS: JMR, LAS

GRANT NUMBER: TED2021-130398B-C21

FUNDER: Spanish Ministry of Science and Innovation

URL FUNDER: https://www.ciencia.gob.es/

INITIALS: JMR, LAS

GRANT NUMBER: PID2022-136522OB-C21

FUNDER: Spanish Ministry of Science and Innovation

URL FUNDER: https://www.ciencia.gob.es/

INITIALS: LAS

GRANT NUMBER: RAICES (IMIENS 2022)

FUNDER: IMIENS (Instituto Mixto de Investigación-Escuela Nacional de Sanidad)

URL FUNDER: " ext-link-type="uri" xlink:type="simple">https://www.imiens.es/"

7. We notice that your supplementary [figures/tables] are included in the manuscript file. Please remove them and upload them with the file type 'Supporting Information'. Please ensure that each Supporting Information file has a legend listed in the manuscript after the references list.

**Additional Editor Comments:**

Dear authors,

thank you for giving us the opportunity to read such an interesting article.

As you can see, the reviewers have raised many concerns, especially of a methodological nature, to which I invite you to pay attention.

The article in its current version needs substantial revision

Reviewers' comments:

Reviewer's Responses to Questions

**Comments to the Author**

1. Is the manuscript technically sound, and do the data support the conclusions?

Reviewer #1: Partly

Reviewer #2: Partly

2. Has the statistical analysis been performed appropriately and rigorously? 

Reviewer #1: Yes

Reviewer #2: No

3. Have the authors made all data underlying the findings in their manuscript fully available?

Reviewer #1: No

Reviewer #2: Yes

4. Is the manuscript presented in an intelligible fashion and written in standard English?

Reviewer #1: Yes

Reviewer #2: Yes

5. Review Comments to the Author

Reviewer #1: In this paper, the authors tackle the issue of fake news detection in the health domain by proposing a pipeline to automatically harvest health-related news and annotate them at the sentence level for Check-Worthiness (CW) and Fact-Checking (FC) in order to perform fake news detection. In order to train and evaluate their approach, a novel dataset is presented, namely KEANE, containing 327 articles belonging to the health-domain and collected from Snopes and Politifact, annotated both at the sentence and at the document level. Main contribution of this work is to train and evaluate on this corpus a fake news detection system that performs CW in order to filter relevant sentences and FC in order to spot sentences containing false claims. For CW, an uncased BERT model trained on the general domain proves to be the most efficient model. For FC, the best results are obtained with an ensemble model that combines three features: the text of the sentence, a subject-predicate-object triple extracted from the sentence, and a sequence of CUI identifiers extracted by mapping sentence tokens to the UMLS terminology. Despite being outperformed by transformers trained on the biomedical domain, this approach shows competitive performances even on unseen data. Moreover, the authors claim that this drawback is compensated by the fact that the proposed method offers more explainability since it identifies the specific sentences in a news article which support false claims.

Main strengths of this paper include a clear motivation on why sentence-level FC is relevant, solid grounding on the literature about fake news detection and the level of detail used to explain both the data collection process and the classification approach. Moreover, the methodology for the data collection is novel and the annotation practices are soundly justified and clearly explained. The approach, albeit its complexity, is presented in detail and reproducible, assuming that the authors provide the data and the documentation on the source code publicly. However, there are significant weaknesses in the manuscript. Sometimes the research is not presented in a scientifically thorough way. Significant details that justify certain methodological choices are lacking and the results, as they are presented, do not fully support the statements of the paper. Moreover, the authors claim that the data will not be released without restrictions, which is not compliant with the PLOS Data Policy. My suggestion for the authors is to publish the data with an unrestricted license on a repository such as Zenodo, which provides a citable DOI and ensures long-term preservation. My final advice is that the article should be accepted under major revisions, upon the publication of the data and after addressing the following points of criticism.

Major points:

- In the section on data collection (Section 4.1), the authors claim to have collected the articles from Snopes and Politifact by looking for those labelled with the category “medical”, “health-check” and “coronavirus”. However, the distribution of articles per category is not presented, raising the issue of whether the corpus is unbalanced and therefore not representative of the whole health domain. For example, the corpus may contain mostly documents related to COVID-19. Please provide the statistics in this section.

- For triple extraction, the authors state that a set of verbs was manually selected (lines 608-612) in order to parse the sentences. However, there is no justification on the criteria used for this selection. It would be explanatory for the reader to see a list of these verbs in the Appendix or at least to understand the rationale behind the selection.

- One of the major issues is the fact that in Section 5.3, in Table 13, only results obtained by applying FC on sentences manually annotated as relevant are presented. This overshadows the fact that the pipeline should be applied by first automatically classifying the sentences as worthy (CW). In fact, as tackled in the Discussion section, the application of CW in tandem with FC can cause a cascade of errors which has an impact on the final results of fake news detection. As a consequence, it would be appropriate to put the results of the whole pipeline on KEANE in order to support the statements of this paper.

- The paper lacks a thorough error analysis to investigate the validity of the results obtained. This section should discuss for which reason sentences which contain facts are not considered worthy and cases of predicted false worthiness. Moreover, it would be interesting to analyse the cases in which the FC model produced incorrect predictions in order to investigate which typology of sentences raise an issue for the model and mitigate the risk of bias.

- As Figure 5 shows, many words can be mapped to wrong CUI terms. Tackling this issue in the discussion is necessary to give enough scientific validity to the algorithm used. Moreover, a way to evaluate the CUI mapping should be proposed as future work.

Minor points:

- In the Introduction, the statistics used to justify the need for fake news detection in the health domain are based on an article published in 2013. It would make the motivation stronger to use more up-to-date data.

- Line 58 starts with “Paper outline”. However, this is probably a typo.

- On line 350-351 and lines 369-371, the authors claim that the results on news detection are high enough to automatically annotate new articles; however, I advise to make this assertion less strong since the model does not achieve enough accuracy to operate autonomously.

- On Table 2, there is a typo on the final news item count. Why is it 327 and not 227? Does this mean that 100 articles in the corpus cannot be evaluated?

- There is an inconsistency for the label FRC for multi-label CW. In Section 2.1, it is stated that a sentence is labelled as FRC if it contains more than a fact. Instead, on lines 493-494 it is written that sentences with facts too complex to be analysed are labelled as FRC. Making this definition consistent throughout the paper will improve its readability.

Reviewer #2: The manuscript's objectives are 3-fold:

1) propose a framework for collecting health-related articles for the tasks of check-worthiness and fact-checking;

2) generate a corpus from the collected articles;

3) design a neural model to classify the articles.

The framework uses a classic text classification pipeline, offering overall very little novelty.

There are a lot of works published that propose how to use word embeddings [1], document embeddings [2], transformers [3], and sentence transformers [4] for the task of misinformation detection. Also, the current literature discusses how network information can improve the detection task [5]. How is this work compared with the proposed model?

Also, why any of the following were not used: word embeddings, document embeddings, or sentence transformers?

After detecting that a piece of information is fake, what should we do with this information? Leave it as it is? There is a large body of work that discusses how we can mitigate the spread of fake news using network immunization, such as proactive approaches [6], tree-based approaches [7], community-based approaches [8], or real-time approaches [9]. I consider this to be very important both in the discussion section and for future work.

The annotation process seems mostly automatic, only at the end does a user check it.

For the Check-worthy annotation, I find it superficial to annotate only 100 sentences.

For the Fact-checking annotation, it is not clearly explained how the manual annotation was done.

For the CheckThat! 2022 results (table 6), the individual papers should also be cited in the table, not only the paper that presents the overview of the challenge.

The results do not seem very promising. I find the scores very low compared with the current literature. Why is this happening?

The experiments seem shallow:

1. There is no hyperparameter tuning evaluation for the models employed. The authors only use the "most promising" hyperparameters

2. Are the results obtained using cross-validation? How many training iterations were used? What are the mean and the standard deviation obtained for each evaluation metric on the test set?

3. There is no time performance evaluation.

4. Ablation testing is missing.

I highly recommend that the GitHub repository be populated for reproducibility purposes.

Also, the code is useless without the collected dataset. Please offer a publicly available repository with the proposed dataset.

Please do a thorough spell-checking of the article before resubmitting.

[1] https://scholar.google.com/scholar?q=misinformation+detection+word+embeddings

[2] https://scholar.google.com/scholar?q=fake+news+document+embeddings

[3] https://scholar.google.com/scholar?q=transformers+misinformation

[4] https://scholar.google.com/scholar?q=fake+news+sentence+transformers

[5] deep+neural+network+ensemble"+social+context" ext-link-type="uri" xlink:type="simple">https://scholar.google.com/scholar?q=fake+news+"deep+neural+network+ensemble"+social+context

[6] fake+news"+"social+network+immunization"" ext-link-type="uri" xlink:type="simple">https://scholar.google.com/scholar?q="fake+news"+"social+network+immunization"

[7] https://scholar.google.com/scholar?q=fake+news+social+media+tree+algorithm+mitigation

[8] https://scholar.google.com/scholar?q=fake+news+network+immunization+community+detection

[9] real+time"+"distributed+system"+"misinformation+detection"+"community+detection"" ext-link-type="uri" xlink:type="simple">https://scholar.google.com/scholar?q="real+time"+"distributed+system"+"misinformation+detection"+"community+detection"

6. PLOS authors have the option to publish the peer review history of their article (what does this mean?). If published, this will include your full peer review and any attached files.

Reviewer #1: **Yes: **Cristian Santini

Reviewer #2: No

---

## [Author Response · Author response to Decision Letter 0]

14 Mar 2024

The response to the reviewers' and editor's comments has been included in the submission as a pdf document called "response_to_reviewers.pdf".

---

## [Decision Letter · Decision Letter 1]

15 May 2024

PONE-D-23-40818R1Building a framework for fake news detection in the health domainPLOS ONE

Dear Dr. Martinez-Rico,

Thank you for submitting your manuscript to PLOS ONE. After careful consideration, we feel that it has merit but does not fully meet PLOS ONE’s publication criteria as it currently stands. Therefore, we invite you to submit a revised version of the manuscript that addresses the points raised during the review process.

Dear Authors, 

I have just received the second reviewer's revisions, which I fully agree with regarding the need to cite some work on word embedding, transformers, and document embedding to detect false information, so that your paper takes this relevant literature into account. 

We look forward to receiving your revised manuscript.

Kind regards,

Ramona Bongelli, Ph.D.

Academic Editor

PLOS ONE

Journal Requirements:

Additional Editor Comments:

Dear Authors,

I have just received the second reviewer's revisions, which I fully agree with regarding the need to cite some work on word embedding, transformers, and document embedding to detect false information, so that your paper takes this relevant literature into account.

Thank you very much.

Good work

Reviewers' comments:

Reviewer's Responses to Questions

**Comments to the Author**

1. If the authors have adequately addressed your comments raised in a previous round of review and you feel that this manuscript is now acceptable for publication, you may indicate that here to bypass the “Comments to the Author” section, enter your conflict of interest statement in the “Confidential to Editor” section, and submit your "Accept" recommendation.

Reviewer #1: All comments have been addressed

Reviewer #2: (No Response)

2. Is the manuscript technically sound, and do the data support the conclusions?

Reviewer #1: Yes

Reviewer #2: Yes

3. Has the statistical analysis been performed appropriately and rigorously? 

Reviewer #1: Yes

Reviewer #2: Yes

4. Have the authors made all data underlying the findings in their manuscript fully available?

Reviewer #1: Yes

Reviewer #2: Yes

5. Is the manuscript presented in an intelligible fashion and written in standard English?

Reviewer #1: Yes

Reviewer #2: Yes

6. Review Comments to the Author

Reviewer #1: The authors have responded with sufficient clarity to all the critical issues in the manuscript and I see no impediment to the publication of the article as it is now presented.

Reviewer #2: Although the authors answered most of my comments, I still think that the manuscript is missing a large chunk of literature regarding word embeddings [1], transformers [2], and document embeddings [4] for detecting fake information. I recommend that the authors mention some of these research endeavors in their related work section so the study is complete, otherwise, it would seem that a large chunk of work was ignored.

[1] https://scholar.google.com/scholar?q=word+embeddings+misinformation+detection

[2] https://scholar.google.com/scholar?q=transformers+misinformation

[3] https://scholar.google.com/scholar?q=fake+news+document+embeddings

7. PLOS authors have the option to publish the peer review history of their article (what does this mean?). If published, this will include your full peer review and any attached files.

Reviewer #1: **Yes: **Cristian Santini

Reviewer #2: No

---

## [Author Response · Author response to Decision Letter 1]

20 May 2024

Dear editor:

We submit the second revision of the original research manuscript entitled “Building a Framework for Fake News Detection in the Health Domain” by Juan R. Martinez-Rico, Lourdes Araujo, and Juan Martinez-Romo, (PONE-

D-23-40818R1) submitted to the PLOS ONE journal.

The authors thank the editor and reviewers for their comments which greatly helped them to improve the contents of the paper. We have prepared a revised version of the paper according to their suggestions, having modified the

article by expanding the “Related work” section with references to works on embeddings and Transformers applied to the detection of disinformation.

Changes and new parts appear in red in this version.

We have also prepared a document with detailed responses to the comments and suggestions of the editor and reviewers.

Thank you for your consideration. We look forward to hearing from you.

---

## [Editor Report · Decision Letter 2]

29 May 2024

Building a framework for fake news detection in the health domain

PONE-D-23-40818R2

Dear Dr. Martinez-Rico,

We’re pleased to inform you that your manuscript has been judged scientifically suitable for publication and will be formally accepted for publication once it meets all outstanding technical requirements.

Kind regards,

Ramona Bongelli, Ph.D.

Academic Editor

PLOS ONE
---

## [Editor Report · Acceptance letter]

4 Jun 2024

PONE-D-23-40818R2 

PLOS ONE

Dear Dr. Martinez-Rico, 

I'm pleased to inform you that your manuscript has been deemed suitable for publication in PLOS ONE. Congratulations! Your manuscript is now being handed over to our production team.

Kind regards, 

on behalf of

Professor Ramona Bongelli 

Academic Editor

PLOS ONE